# Natural variations at the *Stay-Green* gene promoter control lifespan and yield in rice cultivars

Dongjin Shin [1,8], Sichul Lee [2,8 ✉], Tae-Heon Kim [1,8], Jong-Hee Lee[1], Joonheum Park[2], Jinwon Lee[2], Ji Yoon Lee[1], Lae-Hyeon Cho[3], Jae Young Choi [4], Wonhee Lee[5], Ji-Hwan Park[2], Dae-Woo Lee [3], Hisashi Ito[6], Dae Heon Kim [7], Ayumi Tanaka[6], Jun-Hyeon Cho[1], You-Chun Song[1], Daehee Hwang[2,5], Michael D. Purugganan [4], Jong-Seong Jeon [3], Gynheung An[3] & Hong Gil Nam [2,5 ✉]

Increased grain yield will be critical to meet the growing demand for food, and could be achieved by delaying crop senescence. Here, via quantitative trait locus (QTL) mapping, we uncover the genetic basis underlying distinct life cycles and senescence patterns of two rice subspecies, *indica* and *japonica*. Promoter variations in the *Stay-Green* (*OsSGR*) gene encoding the chlorophyll-degrading $Mg^{++}$-dechelatase were found to trigger higher and earlier induction of *OsSGR* in *indica*, which accelerated senescence of *indica* rice cultivars. The *indica*-type promoter is present in a progenitor subspecies *O. nivara* and thus was acquired early during the evolution of rapid cycling trait in rice subspecies. *Japonica OsSGR* alleles introgressed into *indica*-type cultivars in Korean rice fields lead to delayed senescence, with increased grain yield and enhanced photosynthetic competence. Taken together, these data establish that naturally occurring *OsSGR* promoter and related lifespan variations can be exploited in breeding programs to augment rice yield.

[1] Department of Southern Area Crop Science, National Institute of Crop Science (NICS), RDA, Miryang, Republic of Korea. [2] Center for Plant Aging Research, Institute for Basic Science (IBS), Daegu, Republic of Korea. [3] Crop Biotech Institute and Graduate School of Biotechnology, Kyung Hee University, Yongin, Republic of Korea. [4] Center for Genomics and Systems Biology, Department of Biology, New York University, New York, NY, USA. [5] Department of New Biology, DGIST, Daegu, Republic of Korea. [6] Institute of Low Temperature Science, Hokkaido University, Sapporo, Japan. [7] Department of Biology, Sunchon National University, Sunchon, Republic of Korea. [8]These authors contributed equally: Dongjin Shin, Sichul Lee, Tae-Heon Kim ✉email: sciron@ibs.re.kr; nam@dgist.ac.kr

The world's population is expected to increase by 35% over the next 30 years, so crop production must also increase to meet the growing demand[1]. Rice is a staple food for half of the world's population. Although rice-yield potential has increased considerably over the past five decades, mainly through the utilization of semi-dwarf varieties and heterosis[2], it has stagnated worldwide in recent years[3].

Asian rice cultivars belong mostly to two subspecies, *O. sativa* L. ssp. *japonica* and *indica*, which bear distinct morphological and physiological features[4] and show drastically different lifespans, with *indica* showing early senescence[5]. Compared with *japonica*, *indica* exhibits earlier senescence in both whole plants and leaf organs, resulting in a rapid life cycle[6]. *Indica* rice is thought to have evolved under *r*-selection in the tropical zone, emphasizing a rapid cycling life history strategy with a trade-off between rapid reproduction and parental survivorship. The accelerated senescence along with the short lifespan of *indica* rice increases its reproductive output, providing an adaptive strategy in the given environments[7]. However, in temperate and subtropics rice fields, where human population density is high, early senescence of rice cultivars is undesirable because it often results in shorter times for grain filling and poor productivity. Although these two rice subspecies have been used extensively for rice breeding in East, South, and Southeast Asia to improve rice yield and other agronomic traits[8], the effect of senescence has rarely been considered during the breeding process.

Senescence, the final stage of development, is an active phase of orderly degradation and remobilization processes involving a series of changes at the cellular, tissue, organ, and organism levels[9,10]. Senescence affects grain yield and quality in crop species[9,11]. Heritable delays in senescence extend the photosynthetic period and are thus important considerations for improving crop yield[9,12]. Indeed, progressive increases in grain yield in maize and sorghum positively correlate with impaired chlorophyll catabolism, so-called stay-green traits[9,12]. For example, *sgr* mutants in rice and *Arabidopsis* exhibit a stay-green phenotype, due to defects in the chlorophyll-degrading enzyme $Mg^{++}$ dechelatase, resulting in a stay-green phenotype[9,13,14]. However, rice *ossgr* mutants did not maintain photosynthetic capability during the grain-filling period and did not show a yield advantage, indicating that they are nonfunctional *stay-green* mutants[13,15]. Chlorophyll breakdown pathways operating during leaf senescence have been functionally characterized by studying loss-of-function mutants in rice[10]. In addition, a recent genome wide association study with natural variations of rice uncovered new alleles for chlorophyll content and stay-green traits in *japonica* and *indica* rice cultivars[16].

Here, QTL mapping was used to determine the genetic factors responsible for the differential senescence patterns and lifespans between *indica* and *japonica*-type cultivars, and to identify promoter polymorphisms in the *OsSGR* gene. Promoter variations in *indica* lead to higher and earlier induction of *OsSGR*, resulting in accelerated senescence and shorter lifespans. Evolutionary and population genetic analyses show the *indica*-type promoter was acquired from the progenitor subspecies *O. nivara* during its evolution. Lastly, *japonica* OsSGR alleles introgressed into elite *indica*-type cultivars show delayed senescence, longer maintenance of photosynthetic competence, and improved grain yield. These results highlight that *OsSGR* promoter variations can be utilized to improve rice productivity.

## Results

### Age-associated senescence in *japonica* and *indica* cultivars.
The senescence phenotypes of two *japonica*-type cultivars, Junam (JN) and Nampyeong (NP), and two *indica*-type cultivars, IR72 and

IR64, were examined. These cultivars have similar heading dates (Supplementary Fig. 1), and thus reproductive differences are not likely to explain any differences in senescence. The plants were grown in a natural rice field in the National Institute of Crop Science, Korea (35.3° N; 128.5° E), and senescence was monitored by leaf color changes, for instance from green to yellow due to the loss of chlorophyll, and from yellow to brown due partly to oxidation of phenolic compounds in dying cells[10]. The two cultivar types showed clear differences in senescence patterns for whole plants and leaves (Fig. 1a–c), as previously reported[5].

The senescence patterns of individual leaves and panicles were then quantitatively analyzed (Fig. 1b–i). The senescence patterns of the last two leaves are known to critically influence grain yield in rice[17]. Interestingly, these two leaves have distinct senescence-related transcriptome patterns, and show distinct contributions to grain filling[18]. Age-dependent changes of leaf color (Fig. 1b, c), as well as quantification of leaf color changes (Fig. 1d, e) and total chlorophyll levels (Fig. 1f, g), revealed faster senescence in both flag and second upper leaves of the two *indica*-type rice cultivars compared with those of the two *japonica* cultivars. In addition, the second upper leaves showed faster senescence than the flag leaf in both *indica* and *japonica* cultivars. The panicle is the last organ to develop in the aboveground part of the rice, therefore panicle senescence associated with grain maturation determines the aboveground lifespan of whole plants. The panicles of the two *indica* cultivars showed earlier senescence than those of the two *japonica* cultivars (Fig. 1h), as quantified by colourimetric assays (Fig. 1i).

### QTL mapping and map-based cloning of *OsSGR*.
To determine the genetic loci responsible for the different senescence patterns between the two subspecies, QTL mapping was performed with $F_{2:3}$ populations derived from a cross between the *indica* cultivar IR72 and the *japonica* cultivar JN. Leaf senescence phenotypes in the $F_{2:3}$ population showed a continuous variation, exhibiting the characteristics of quantitative inheritance (Supplementary Fig. 2a, b). The senescence-associated QTL of the flag and second upper leaves mapped separately, consistent with their different senescence schedules (Fig. 1b–g). The senescence differences between the two parental cultivars were governed by multiple loci (Fig. 2a, b; Supplementary Table 1). Three loci above the 95% confidence cutoff were shared by the flag and second upper leaves in the two cultivars, whereas three others distinctly controlled senescence in the flag and second upper leaves.

The genetic locus on chromosome 9 with the highest LOD score was examined to identify the gene(s) responsible for the differential senescence phenotypes. The locus was associated with early senescence of both the flag and second upper leaves, suggesting that it influences senescence of the whole plant rather than a specific leaf. 6,349 $BC_5F_2$ lines derived from a cross between *japonica* JN and *indica* IR72 were generated and used for fine mapping of the early-senescence locus, with molecular markers developed for the *indica* and *japonica* subspecies (Supplementary Table 2). The locus for the *indica*-type early leaf senescence was detected between markers C9–10 and C9–12, which spans ~26 kb of chromosome 9 (Fig. 2c; Supplementary Fig. 3). This region was examined for known coding sequence(s) that could underlie early leaf senescence, revealing *OsSGR* (LOC_Os09g36200), *Harpin-induced protein 1* (LOC_Os09g36210), and *OsPRR95* (LOC_Os09g36220). *OsSGR* encodes a $Mg^{++}$-dechelatase that is involved in chlorophyll degradation, which promotes degradation of the light-harvesting complexes and senescence of leaves and whole plants[13,15,19,20]. *OsPRR95*, a two-component signaling gene[21], is a homolog of *Arabidopsis PRR9*, which controls age-dependent leaf

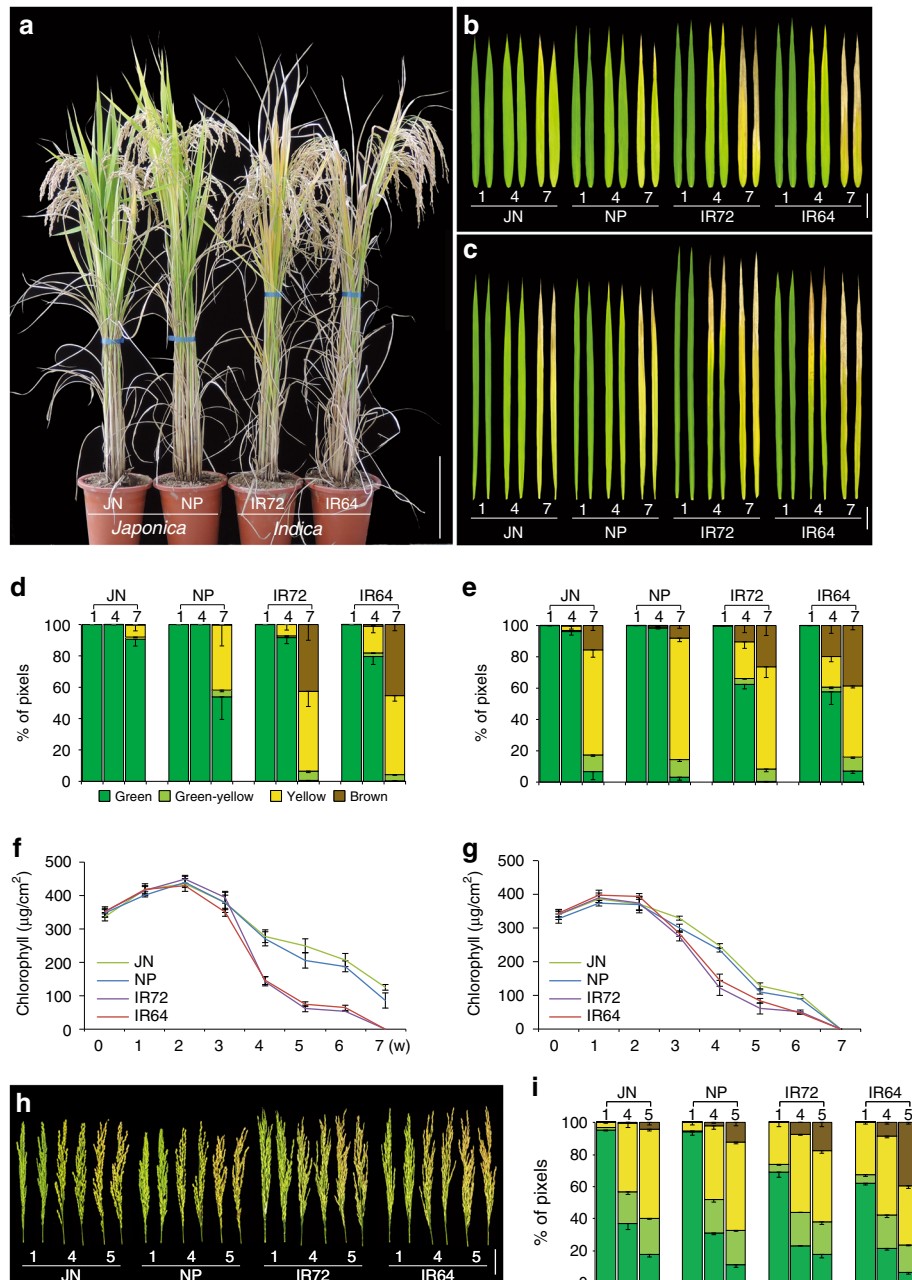

**Fig. 1 Age-associated leaf senescence in *japonica* and *indica*-type rice plants. a** Representative senescence phenotypes of two *japonica* (JN, Junam; NP, Nampyeong) and *indica* (IR72, IR64) type rice plants grown in natural field conditions. Scale bar, 20 cm. Color changes of the flag (**b**) and second upper (**c**) leaves during seasonal senescence (1, 4, and 7 weeks after heading). Bars, 5 cm. Quantification of color changes of the flag (**d**) and second upper (**e**) leaves at 1, 4, and 7 weeks after heading by Automated Colourimetric Assay. The individual flag and second leaves were categorized into four groups according to their color: green, green-yellow, yellow, and brown. The percentage of each group is presented. Temporal changes of chlorophyll levels in the flag (**f**) and second upper (**g**) leaves from 0 to 7 weeks after heading. Color changes (**h**) and quantification of color (**i**) of panicles from 1, 4, and 5 weeks after heading. Data are means ± SE; *n* ≥ 3 independent samples. Source data of Fig. 1d–g and i are provided as a Source Data file.

senescence[22], and *Harpin-induced protein 1* has been implicated in stress responses such as disease resistance[23].

Activation-tagging lines for *OsSGR* and *OsPRR95* were isolated from an activation-tagging library generated in a *japonica* cultivar Dongjin, and their senescence patterns were examined. Leaf and whole-plant senescence were unaltered in activation lines of *OsPRR95* compared to controls (Supplementary Fig. 4), but were accelerated in two activation lines of *OsSGR* (Supplementary Fig. 5a–g). *OsNAP*[24], a senescence marker gene, was more highly expressed in these mutants than in WT (Supplementary Fig. 5d). In addition, chlorophyll contents in a whole plant and

leaves were maintained higher in *ossgr* mutants of *japonica* cultivars (Supplementary Fig. 6a–f), consistent with previous reports[13,15,25]. Leaf senescence, as probed by *OsNAP* expression, proceeded in these mutants but appeared to be partially delayed compared to wild-type plants. Chlorophyll loss in panicles was also accelerated in *OsSGR*-activation lines, and delayed in *ossgr* mutants (Supplementary Figs. 5h, i and 6g, h). These results suggest that, among the three genes in the 26-kb region, *OsSGR* is responsible for early senescence of *indica* rice plants, leading to the rapid life cycle. Activation of *OsSGR* led to reduced grain yield with poor agronomic traits (Supplementary Fig. 5j–n), but *ossgr*

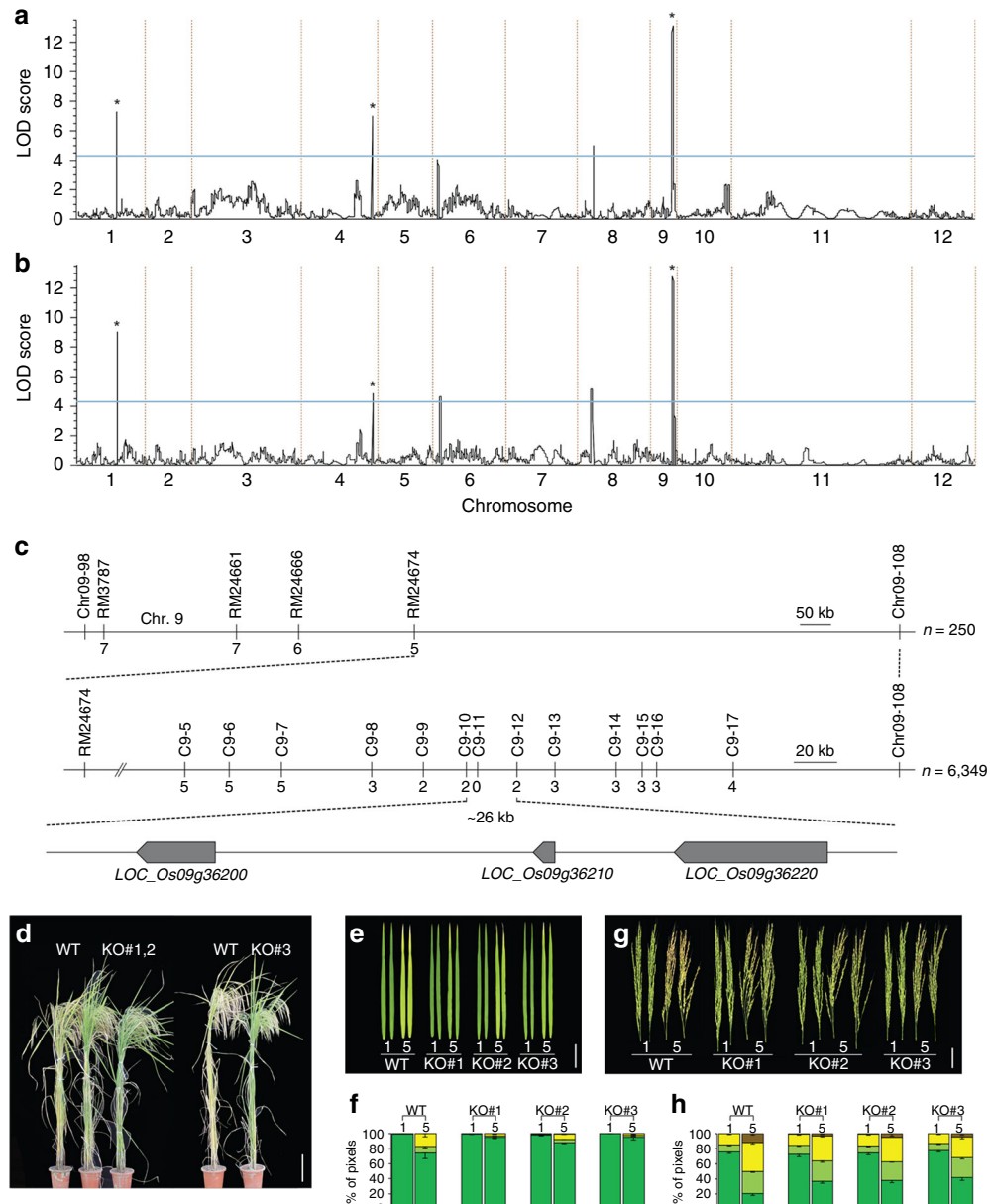

**Fig. 2 OsSGR is responsible for the differential leaf senescence.** Mapping curves of QTLs controlling the senescence of the flag (**a**) and second upper (**b**) leaves on the 12 rice chromosomes via GBS (genotyping-by sequencing). A LOD (log likelihood) significance threshold of 4.3 (blue lines) was used to declare the presence of significant QTL in a genomic region. Three common genetic loci between two leaves are indicated as *. **c** Fine mapping of the locus on the chromosome 9 for the leaf senescence with molecular markers. The 26-kb genomic region between C9-10 and C9-12 markers contains three predicted coding sequences (thick arrow). **d** Senescence phenotypes of WT (*indica* cultivar Kasalath) and mutants (KO#1, 2, and 3) generated by CRISPR/ Cas9 editing. Scale bar, 20 cm. Color changes (**e**, **g**) and quantification (**f**, **h**) of the flag leaves (**e**, **f**) and panicles (**g**, **h**) of WT and the *indica ossgr* mutants during seasonal senescence (1 and 5 weeks after heading). Scale bars, 5 cm. Values are means ± SE ($n \geq 3$). In colourimetric assay, the same scale is used as in Fig. 1. Source data of Fig. 2f and h are provided as a Source Data file.

mutants did not show any yield advantage (Supplementary Fig. 6i–m).

To verify that the *indica* allele of *OsSGR* is required for its early-senescence phenotype, *OsSGR* knockout (KO) mutants were generated in the *indica* cultivar Kasalath via CRISPR/Cas9 genome editing (Supplementary Fig. 7a). The KO *indica* plants showed delayed loss of chlorophyll in whole plants, leaves, and panicles (Fig. 2d–h; Supplementary Fig. 7b). Furthermore, RNAi-mediated silencing of *OsSGR* in the *indica* cultivar led to delayed chlorophyll loss (Supplementary Fig. 8). Leaf senescence, as probed by *OsNAP* expression, also proceeded in these mutants but appeared to be partially delayed compared with their parental

lines (Supplementary Figs. 7c and 8e). These observations suggest that the *indica* allele of *OsSGR* contributes to its relatively early senescence, short lifespan, and rapid life cycle.

To determine how the *indica* allele of *OsSGR* leads to earlier senescence, genomic polymorphisms between the *OsSGR* alleles of the parental mapping cultivars JN (*japonica*) and IR72 (*indica*) were examined. There were four nonsynonymous SNPs in the coding region, and 13 SNPs and 3 indels in the promoter region (Supplementary Fig. 9). As *OsSGR* encodes a $Mg^{++}$-dechelatase enzyme that converts chlorophyll a to pheophytin a[19], the amino acid changes in the coding region of *OsSGR* were first tested for their effect on senescence. Transgenic *japonica* JN rice plants that

ubiquitously overexpressed either the *indica* or *japonica* OsSGR allele were generated and found to display similar early leaf senescence phenotypes (Supplementary Fig. 10), suggesting that the SNPs in the coding region of the two alleles are not responsible for the early *indica*-type leaf senescence (Supplementary Fig. 10c, d). In addition, the two alleles displayed similar chlorophyll-degradation activities in vitro (Supplementary Fig. 11a) and *in planta* (Supplementary Fig. 11b). The *ossgr* allele did not show any chlorophyll breakdown activity (Supplementary Fig. 11a, b).

**OsSGR promoter polymorphisms are associated with senescence.** OsSGR expression levels during senescence were examined to determine whether the OsSGR promoter polymorphisms contributed to the differential senescence (Supplementary Fig. 9). OsSGR was induced in an age-dependent manner during senescence in both *japonica* cultivars and both *indica* cultivars (Fig. 3a). Importantly, OsSGR expression was higher and earlier in the two *indica* cultivars than in the two *japonica* cultivars (Fig. 3a). Furthermore, a transient assay in rice protoplasts revealed that OsSGR promoter segments from the *indica* cultivar IR72 had 3.1-fold higher activity than those from the *japonica* cultivar JN (Supplementary Fig. 12). This result suggests that higher expression of OsSGR in *indica* rice is responsible for its early-senescence phenotype.

To further investigate the role of OsSGR promoter polymorphisms in senescence, an association analysis was performed between the OsSGR promoter region and leaf senescence phenotypes in 105 rice accessions belonging to five *O. sativa* subgroups[26] (Supplementary Fig. 13 and Supplementary Data 1). Sequencing of the OsSGR promoters of these accessions identified 21 polymorphic sites that establish eight haplotypes (Fig. 3b). Each haplotype showed distinct chlorophyll levels, with haplotype 1 (*indica*) showing substantially lower levels than haplotype 8 (*japonica*) (Fig. 3c). A comparison of chlorophyll levels between the two groups of accessions with and without the polymorphism at each site showed that 16 of the 21 polymorphic sites were significantly associated with chlorophyll content (Fig. 3d). Furthermore, among the 16 sites, 15 were associated with a difference in chlorophyll content between *indica* and *japonica* rice plants. Together, these data strongly suggest that OsSGR promoter polymorphisms broadly influence senescence, lifespan, and life cycle in various rice accessions.

Phylogenetic analysis of the OsSGR promoter sequences in the 105 rice accessions with wild rice sequences indicated that *indica* groups with *O. nivara*, whereas *japonica* groups with *O. rufipogon* (Fig. 3e). *O. nivara* is considered to be the progenitor of *indica*, whereas *O. rufipogon* is regarded to be the progenitor of *japonica*[26]. Phylogenetic analysis of an expanded set of wild and domesticated rice samples further supported this relationship (Supplementary Fig. 14). A group of *O. rufipogon*-like wild rice is considered to be a sister group to all domesticated rice and its progenitor wild rice (Supplementary Fig. 14)[27]. Therefore, the *indica*-type promoter variations may be derived from *japonica* or from the progenitor of *O. rufipogon*. Efficient seed production requires coincidence of panicle development with leaf senescence that leads to generation of the nutritional resources for grain filling. The *indica*-type promoter of OsSGR could have played a critical role during the evolution and domestication of *indica* type for a rapid life cycle, through concomitant control of leaf and panicle senescence. There was no evidence of a selective sweep in the region surrounding OsSGR (Supplementary Fig. 15), suggesting the promoter variations of OsSGR in *indica* rice may have existed in *O. nivara* and were later inherited into *indica*.

**Introgression of *japonica* OsSGR allele into *indica*.** Three elite rice cultivars that are bred in Korea (Milyang21 and Milyang23) and in IRRI (IR72) have *indica* OsSGR promoter alleles with haplotype 1 (Supplementary Table 3a). When grown in a conventional paddy rice field in the National Institute of Crop Science, Korea (35.3° N; 128.5° E), grain yields per plant in the year 2018 were 35.9 (IR72), 29.0 (Milyang21), and 29.1 g plant$^{-1}$ (Milyang23) (Supplementary Table 3b). We hypothesized that these elite cultivars undergo undesirable early senescence due to the presence of the *indica* OsSGR alleles, decreasing the yield due to a reduced grain-filling period. This hypothesis predicts that replacing the *indica* OsSGR alleles in these cultivars with a *japonica* OsSGR allele would increase the yield and grain filling.

To test this hypothesis, near isogenic lines (NILs) of these lines harboring a *japonica* OsSGR allele were generated by crossing them with JN or Saeilmi (haplotype 8). The three NILs harboring the *japonica* OsSGR allele (IR72-NIL, Milyang21-NIL, and Milyang23-NIL) all showed delayed senescence (Fig. 4a–c, g–i; Supplementary Fig. 16a, b) and the expression of OsSGR was reduced in IR72-NIL compared to IR72 (Fig. 4j). The lower expression of OsNAP in IR72-NIL also supported delayed senescence (Supplementary Fig. 17a). To evaluate photosynthetic competence in NILs, the net $CO_2$ assimilation rates and Fv/Fm ratios representing the photosynthetic ability in the flag leaves were measured (Fig. 4k; Supplementary Fig. 17b, c). Three NILs of the *indica* background harboring *japonica* OsSGR allele displayed extended photosynthetic competence with higher chlorophyll contents (Fig. 4g, i, k; Supplementary Fig. 17b, c). The relative growth rates (RGR) quantifies plant growth speed and is considered to be a reliable standard for estimating plant productivity[28,29]. It is calculated as the dry mass increment per aboveground biomass at a given time point. RGRs of IR72-NIL are shown in Supplementary Fig. 17d. The RGR of all the rice plants we examined declined continuously during the grain-filling stage. However, IR72-NIL harboring *japonica* OsSGR allele maintained higher RGR, especially between 5 and 7 weeks after flowering, than the parental IR72 (Supplementary Fig. 17d), indicating the contribution of the *japonica* OsSGR allele in higher biomass productivity.

On the other hand, JN-NIL, generated by replacing the *japonica* OsSGR allele (haplotype 8) in JN with the *indica* OsSGR allele in IR72 (haplotype 1), showed earlier and higher expression of OsSGR and OsNAP during the senescence period along with earlier senescence phenotypes across whole plants, flag leaves, and panicles (Fig. 4d–h; Supplementary Fig. 17e–g). The photosynthetic activity of JN-NIL was lower than JN during the grain-filling period (Supplementary Fig. 17h, i). JN-NIL harboring the *indica* OsSGR allele showed relatively faster decline of the RGR value than its parental cultivar (Supplementary Fig. 17j), indicating the negative effect of the *indica* OsSGR allele in biomass productivity in our field condition.

All the NILs with the *japonica* OsSGR allele showed higher grain yields. The grain yields of the IR72-NIL, Milyang21-NIL, and Milyang23-NIL were 39.6, 32.7, and 32.6 g per plant, which corresponds to increases of 10.6, 12.7, and 12.0%, respectively, compared with their parental cultivars (Fig. 4l; Supplementary Table 3b). The increased grain yield was positively correlated grain-filling rate (Fig. 4m; Supplementary Table 3b) during the grain-filling stage. In contrast, JN-NIL with the *indica* OsSGR allele showed lower grain yields with 27.6 g per plant, which corresponds to decrease of 10.1% (Fig. 4l; Supplementary Table 3b). Furthermore, the grain-filling rate was reduced by 6.5% in JN-NIL plants (Fig. 4m; Supplementary Table 3b) compared with its parental JN plants. The grain yields of the all NIL lines were further confirmed through a large-scale field trial

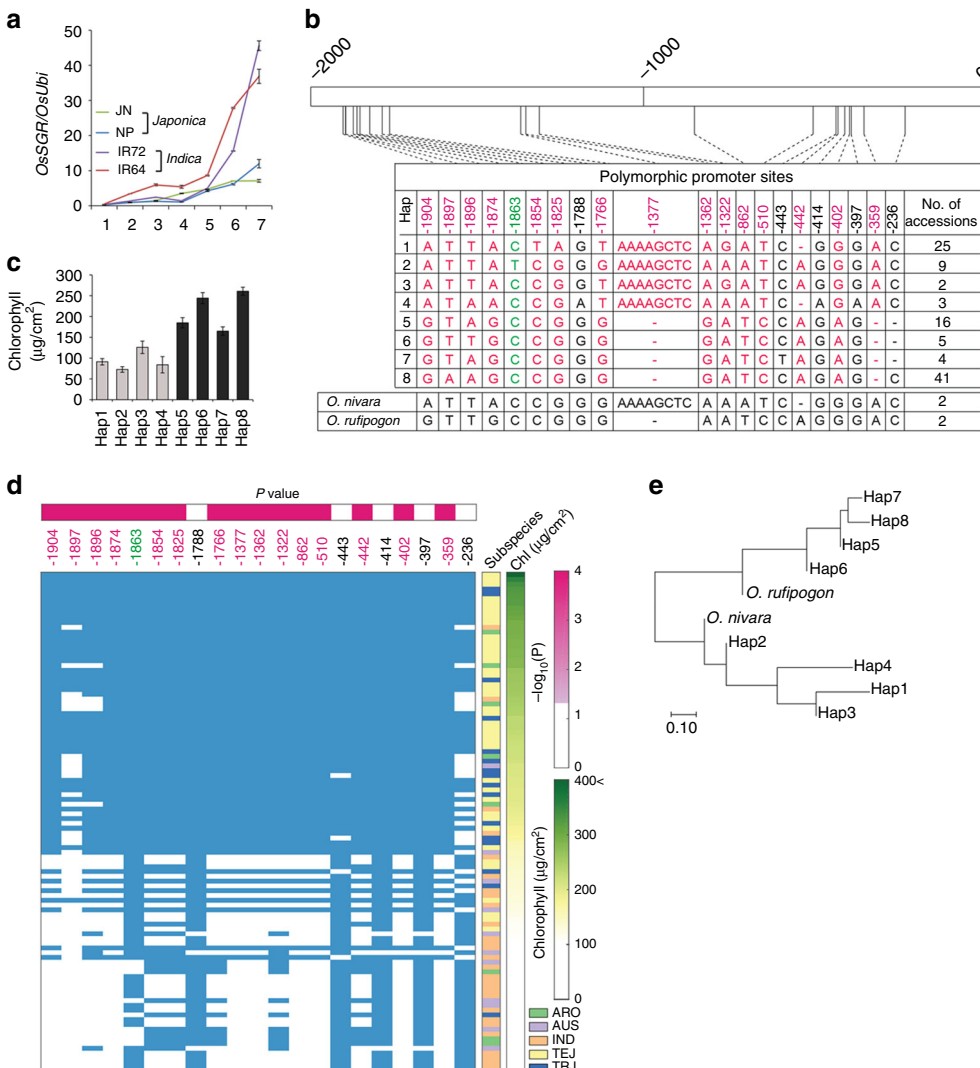

**Fig. 3 Analysis of natural variations in *OsSGR* promoter. a** *OsSGR* transcript levels in the flag leaves of two *japonica* (JN, NP) and two *indica* (IR72, IR64) cultivars measured at seven time points with 1-week intervals from 1 to 7 weeks after heading. Data are means (± SE) of three biological repeats.
**b** Haplotype analysis of the polymorphic *OsSGR* promoter region from 105 accessions. These accessions showed 21 polymorphic sites in the 2-kb promoter region, which split into eight haplotypes (Hap). Note that IR72 and IR64 (*indica*) belong to haplotype 1 and JN and NP (*japonica*) to haplotype 8. Genetic variations of 2 *O. nivara* and 2 *O. rufipogon* accessions were also noted at the bottom. **c** Average chlorophyll concentration of the flag leaves in each of the eight haplotype plants at 6 weeks after heading. Data are means ± SE for each haplotype. **d** Association analysis between the promoter polymorphisms and the chlorophyll levels among the 105 accessions. The accessions (rows) are shown with major subspecies to which they belong (middle panel) and are sorted in descending order of their chlorophyll contents (right panel). For each polymorphism in the promoter (column in the left panel), shown is the significance (upper left panel) of the difference in chlorophyll content between the accessions of which DNA base are the same (blue boxes) and different (white) from those of *japonica*. The 16 polymorphic sites noted in red and green (**b**, **d**) are associated with chlorophyll level. The polymorphic sites in red are same to those between *indica* and *japonica*. **e** Phylogenetic tree of the eight haplotypes (Hap) and the two progenitor subspecies, *O. nivara* and *O. rufipogon*. Promoter alleles of *indica* (Haplotype 1) and *japonica* (Haplotype 8) are grouped with those of *O. nivara* and *O. rufipogon*, respectively. The scale bar represents the number of nucleotide substitutions per nucleotide position. Source data of Fig. 3a and c are provided as a Source Data file.

in the following year. Their grain yields and the senescence phenotypes in the field are shown in Supplementary Fig. 18.

## Discussion
Rice is a daily staple food crop for over 10 billion people, and improving crop productivity is an important task to meet future demands of the growing world population[1]. Senescence is an active phase that involves well-orchestrated degradation and remobilization processes that affect crop productivity and quality[9]. Here, we explore the differential senescence patterns of two rice subspecies, *japonica* and *indica*, as a rice-breeding trait. The two rice subspecies show different senescence patterns and

lifespans, which are regulated through quantitative inheritance (Figs. 1, 2a–b; Supplementary Fig. 2). In this study, promoter variations in the *OsSGR* gene are uncovered as one of the main genetic mechanisms underlying differential senescence between the two subspecies (Fig. 2c; Supplementary Fig. 9). Allelic polymorphisms in the *OsSGR* promoter in *indica* cultivars lead to higher and earlier expression of *OsSGR* and thereby to their early-senescence phenotype (Fig. 3a), affecting the senescence and lifespan traits broadly in various rice accessions (Fig. 3b–d).

When we analyzed the binding motifs for transcriptional factors in the 2-kb *OsSGR* promoter region using NEW PLACE (A Database of Plant Cis-acting Regulatory DNA element; https://

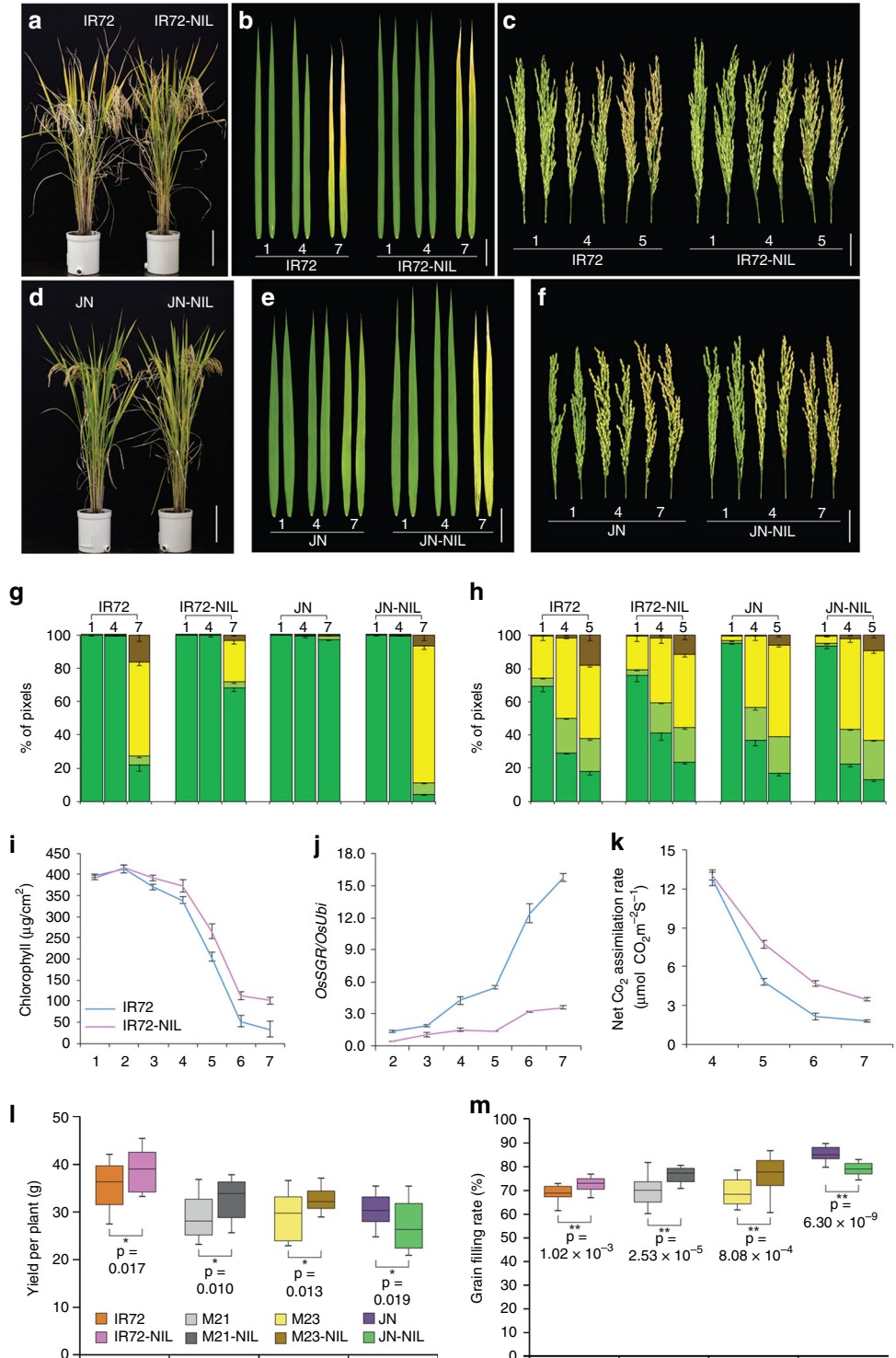

**Fig. 4 The *japonica* allele of *OsSGR* increases grain filling and yield when introgressed into elite *indica* cultivars. a** Whole-plant senescence phenotypes of an *indica*-type cultivar (IR72) and its near isogenic line (IR72-NIL) harboring the *japonica OsSGR* allele grown in paddy field. Scale bar, 20 cm. Color changes of the flag leaves (**b**) and panicles (**c**) from plants in (**a**) during seasonal senescence (1, 4, and 7 weeks after heading). Scale bar, 5 cm. In colourimetric assay, the same scale is used as in Fig. 1. **d** Senescence phenotype of a *japonica* cultivar (JN) and its near isogenic line (JN-NIL) harboring the *indica OsSGR* allele. Scale bar, 20 cm. Color changes of the flag leaves (**e**) and panicles (**f**) from plants in (**d**). Quantification of leaf color change in the flag leaves (**g**) and panicles (**h**). Data are means ± SE (*n* ≥ 4). **i** Temporal changes of chlorophyll levels of flag leaves from 1 to 7 weeks after heading. Data are means ± SE (*n* = 8). **j** Analysis of *OsSGR* expression in the flag leaves from plants in (**a**, **d**) from 2 to 7 weeks after heading. Data are means ± SE (*n* = 3). **k** The analysis of the net $CO_2$ assimilation rate in flag leaves of IR72 and IR72-NIL during the grain-filling stage (4, 5, 6, and 7 weeks after heading). Data are means ± SE (*n* = 5). Total grain yields per plant (**l**) and grain-filling rates (**m**) of the near isogenic lines (NIL). In the box plots, the center value is the median, and the bottom and top edges of the boxes display 1.5 times the interquartile range. Values are the mean ± SE (*n* ≥ 16). *$P < 0.05$, **$P < 0.01$ by Student's *t* test. Source data of Fig. 4g–m are provided as a Source Data file.

www.dna.affrc.go.jp/PLACE/?action=newplace), we found that the *japonica* and *indica* promoters contained same numbers (13) of WRKY-biding motifs and no NAC-binding motif. However, there is a distinctive difference in the Dof-binding motifs between the two promoters. There are 14 Dof protein-binding motifs (consensus AAAG sequences or its reversibly complementary sequence, CTTT[30]) in the *japonica* promoter. On the other hand, in the *indica* promoter, there is a new Dof protein-binding motif formed by insertion of AAAAGCTC (position −1, 377; Supplementary Fig. 9). This region with the new Dof-binding motif is tightly associated with low levels of chlorophyll contents. We anticipate that this new Dof-binding motif, and the respective Dof transcription factor may together lead to the *indica*-type phenotype of chlorophyll loss. Elucidating the molecular mechanism regulating the early and higher induction of *OsSGR* should be a key future effort.

Population genetic and evolutionary analyses revealed that *indica* and *japonica* alleles originate from *O. nivara* and *O. rufipogon* rice, respectively (Fig. 3e; Supplementary Fig. 14), indicating that the *indica*-type promoter of *OsSGR* had a critical role during the evolution and domestication of *indica* rice for a rapid life cycle. We propose that a critical event in the evolution of the *indica* subspecies as a rapid cycling rice is acquisitions of variations in the promoter, rather than the coding sequence, of the chlorophyll-degrading *OsSGR* gene. Senescence, which determines lifespan, is a fundamental question in evolutionary ecology. Here, we provide a crucial evolutionary mechanism of the senescence and death for the *r*-selection life history in rice plants.

The nature of *OsSGR* as a nonfunctional stay-green gene has been well-characterized previously[13,15]. *OsSGR* encodes the enzyme $Mg^{++}$-dechelatase in the chlorophyll-degradation pathway, and is not a regulatory gene[19]. We conducted chlorophyll-degradation activity assays in vitro and in planta to analyze the biochemical activity of the proteins derived from various *OsSGR* alleles (Supplementary Fig. 11a, b). The results showed that the protein from the *ossgr* knockout mutant allele showed a negligible enzyme activity toward chlorophyll degradation. Accordingly, *ossgr* mutant plants maintain a high level of chlorophyll content in senescent leaves but without comparable maintenance of net photosynthesis[15], which is a characteristic feature of nonfunctional stay-green mutants. However, the *ossgr* knockout mutation leads to delay in some aspects of functional senescence upon a prolonged growth; Fv/Fm value was higher in 30 days after heading in the mutant plants than in wild-type plants[15]. This previous observation is consistent with the expression pattern of *OsNAP*[24], a senescence marker gene, in *ossgr* mutant plants. Expression of *OsNAP* is increased in both wild-type and *ossgr* mutant plants during the grain-filling stage, indicating that *ossgr* mutant plants undergo senescence despite of their stay-green phenotype. However, the expression of *OsNAP* was lower in *ossgr* knockout mutants than in wild-type plants, indicating a partial delay of a functional senescence in terms of *OsNAP* expression (Supplementary Fig. 6d, 7c) in addition to the Fv/Fm values[15].

Here, we showed that rice grain yield can be increased by replacing the *indica* allele of *OsSGR* with the *japonica* allele. As *OsSGR* is a nonfunctional stay-green gene in terms of net photosynthesis, it seems paradoxical to increase rice yield via *OsSGR*. It was the difference in the promoter region of the *indica* and *japonica* alleles of the *OsSGR* gene that led to yield increase in the NILs we generated. Unlike the *ossgr* knockout mutation with no enzyme activity of OsSGR, the proteins from *japonica* and *indica* alleles showed comparable enzyme activity (Supplementary Fig. 11). However, the promoter regions of the two alleles of *OsSGR* are diverged (Supplementary Fig. 9). Compared with the promoter of the *japonica* allele, the promoter of the *indica* allele led to earlier and higher induction of *OsSGR* (Fig. 3a) and to

earlier loss of chlorophyll (Fig. 1) with concomitant reduction of photosynthesis (Fig. 4k) and grain yield (Fig. 4l). The senescence response in rice plants with the *indica* allele is similar to that in the activation-tagging lines (Supplementary Fig. 5) with the *japonica* alleles, where *OsSGR* expression is increased at a later stage with lower level of chlorophyll and reduced grain yield (Supplementary Fig. 5). In generating the NIL lines with increased yield, we replaced the *indica* allele with the *japonica* allele, so that the induction of *OsSGR* is slower than the parental lines with the *indica* allele (Fig. 4j). This leads to slower loss of chlorophyll (Fig. 4g. i), higher photosynthesis (Fig. 4k), and increased yield (Fig. 4l) compared with their parental lines. Thus, the senescence response of rice plants with the *japonica* allele with slower induction of *OsSGR* became comparable with a functional stay-green phenotype, unlike the *ossgr* knockout mutant which showed nonfunctional stay-green phenotype with no yield advantage[13,15]. As the heading date of these lines are same, our results show that a senescence period with extended photosynthetic competence can lead to higher productivity. In agreement with this notion, a recent report showed that an extended photosynthetic competence during senescence stage leads to increased harvest index in SGM-3 mutant of upland rice variety Nagina 22; SGM-3 was suggested to be a "novel and functional stay green mutant"[31].

Lifespan indicates the maximal life expectancy from the seed to seed and the length of time that plants live or expected to live[32]. In this regard, the lifespan of rice plants is the duration time from seed germination to the panicle senescence and death associated with grain maturation. In analyzing senescence processes of rice plants in our experiments, we chose two *japonica* and *indica* rice cultivars, which showed the same heading date (Supplementary Fig. 1) to avoid an influence of the differences in reproductive timing on senescence processes and the related lifespan. Despite of the same heading dates, the panicles of the two *indica* cultivars showed earlier senescence than those of the two *japonica* cultivars (Fig. 1h), as quantified by colorimetric assays (Fig. 1i). Thus, the lifespans of these rice cultivars were largely related to senescence process of the panicles after heading, which is controlled by the differential expression levels and induction kinetics of *OsSGR* (Fig. 4c, f, h, j).

Optimal lifespans and senescence patterns of rice varieties have been selected to maximize rice productivity and/or economic income based on their cultivating climates and cropping systems. In the single-cropping region, including most of *japonica* cultivation areas, the balance of grain-filling period and rate is an important agricultural trait for maximum crop production. Thus, the lower transcript levels of *OsSGR* in *japonica* lead to lower chlorophyll degradation and extended photosynthesis capacity, leading to improved productivity with extended grain-filling period and a balanced nutrient remobilization rate in the single-crop system of *japonica*-cultivating areas. On the contrary, most of *indica*-cultivation areas have double- or triple-cropping systems. Therefore, cultivars with a short lifespan and earlier senescence associated with rapid grain-filling rate in a short grain-filling period should be favored for total maximum output for a given year. For this reason, the promoter of *OsSGR* in *indica* was naturally selected for faster and higher expression of *OsSGR*. In fact, promoter variation leads to seasonally higher expression of the *OsSGR* gene in the *indica* subspecies, which in turn triggers earlier senescence of leaves and panicles leading to a rapid life cycle. Thus, the seasonal induction kinetics of the *OsSGR* promoter variations and the related lifespan variations are key ecological and agronomic traits in rice evolution. On the other hand, natural selection on the coding region of *OsSGR* may have been unfavorable during domestication as the continual impairment of enzyme activity might have not been beneficial, or might even be

detrimental for crop productivity, as observed in the nonfunctional *ossgr* mutants, or in the *OsSGR*-overexpression lines with early senescence without an adaptive advantage.

This work shows that introgression of the *japonica OsSGR* allele into elite *indica*-type cultivars delays senescence, thereby further increasing grain filling and yield in the already high-yielding cultivars in the single-cropping systems in Korean rice field. Thus, utilization of the naturally occurring *OsSGR* alleles provides a beneficial breeding strategy in rice.

## Methods

**Plant materials.** For QTL analysis and NIL development, we used three *indica*-type cultivars, IR72, Milyang21 (M21), and Milyang23 (M23), and two *japonica*-type cultivars, Junam (JN) and Saeilmi, showing early- and late-senescence phenotype, respectively. A $F_{2:3}$ populations (141 lines) derived from a cross between IR72 and JN were used to identify leaf senescence phenotype. IR72-NIL (harboring *japonica*-type *OsSGR* allele), M21-NIL, and M23-NIL plants were generated by backcrossing the IR72 × JN, M21 × JN, and M23 × Saeilmi and further four times with its recurrent parents, IR72, M21, and M23, respectively. JN-NIL (harboring *indica*-type *OsSGR* allele) plants were generated by backcrossing the JN × IR72 line and further six times with JN. For haplotype and phylogenetic analysis, the 105 cultivated rice varieties (Supplementary Data 1) including the 30 *indica*, 41 *temperate japonica*, 16 *javanica* (*tropical japonica*), 8 *aromatic*, and 10 *aus*, were obtained from a collection of National Agrobiodiversity Center, RDA, Republic of Korea. Two *O. rufipogon* and two *O. nivara* accessions were obtained from *Oryzabase* and NIAS Genebank, respectively. Accessions used in this study were cultivated in the paddy field located at National Institute of Crop science, RDA (Miryang; 35.3° N; 128.5° E) in 2017 and 2018.

**QTL analysis and gene cloning of the locus on chromosome 9.** The construction of genetic linkage map and QTL analysis were carried out based on the genotypes and leaf chlorophyll contents of $F_{2:3}$ populations derived from a cross between IR72 and JN using the QTL IciMapping program by inclusive composite interval mapping (ICIM) with 1000 permutations (Supplementary Table 4 and Supplementary Data 2)[33,34]. The QTL locus on chromosome 9 was first delimited to a genomic interval between the markers Chr09-99 and Chr09-107 on Chromosome 9 (Supplementary Data 2). To narrow down of locus, one line was selected from $F_{2:3}$ population showing early-senescence phenotype and containing the genomic interval between the markers Chr09-98 and Chr09-108. Furthermore, we crossed this line four times with the *japonica*-type JN as a recurrent and the resultant $BC_4F_1$ plants were selfed to obtain $BC_4F_6$ population. The $BC_4F_6$ population with 250 lines was screened using seven interval markers (Supplementary Table 2). The chlorophyll contents of leaves were investigated as the senescence phenotype. For the gene cloning, two of $BC_4F_6$ lines were chosen and cross these two lines with JN again. To obtain enough recombinants for fine mapping, $BC_5F_2$ with 6349 plants were screened using four flanking markers (RM24674, C9-5, C9-13, and C-17). Then new markers were developed to genotype those recombinants. Finally, the locus on chromosome 9 was delimited to a 26-kb genomic interval between the markers C9–10 and C9–12. For progeny test, we tested the genotype and chlorophyll content of leaf of thirteen recombinant lines to confirm the fine-mapping results using primers listed in Supplementary Table 2.

**Determination of leaf senescence.** Two methods were adopted to evaluate the leaf senescence during grain-filling stage. The chlorophyll contents were estimated once a week by measuring the middle parts of the flag or second upper leaves using a CCM-300 chlorophyll meter (Opti-Sciences, Hudson, NH). To quantify the progressive leaf and panicle color modification from green to green-yellow, to yellow, and to brown, the Automated Colourimetric Assay (ACA) described in the previous report was used[35]. After scanning of detached leaves or panicles, background removed images of single leaf or panicle was prepared by using ImageJ. After that, the percentage of colors (green, green-yellow, yellow, and brown) within the extracted images in a pixel-wise manner by using *R* implemented ACA software was calculated.

**$Mg^{++}$-dechelatase assay of OsSGR.** To compare the biochemical activities of *indica* (IR72)-OsSGR *japonica* (JN)-OsSGR and *ossgr*-type proteins, we examined the $Mg^{++}$-dechelating activity of three mature OsSGR proteins prepared by a wheat-germ protein-expression system, as previously described[19]. Recombinant OsSGR proteins were synthesized with an in vitro transcription/translation system (TNT SP6 High-Yield Wheat Germ Protein Expression System; Promega). Transit peptide was removed, and a FLAG-tag was introduced at the C-terminus of the OsSGR proteins. First, the DNA fragments (5′-TCC CCA CCG CGC GAT <u>AAG CTT</u> GAC TAC AAA GAC GAT GACGAC AAG TGA AAA CGA ATT <u>CGA GCT</u>-3′, the underlined section is *Hind*III site) were cloned into the pF3A WG (BYDV) Flexi vector (Promega) using an In-Fusion cloning system (Clontech Laboratories) to introduce *Hind*III site. Then, the *OsSGR* DNA fragments were amplified using the primer pair OsSGR-WG-F and OsSGR-WG-R (Supplementary

Table 5) and cloned into the pF3A WG *Hind*III site using an In-Fusion cloning system. Plasmid DNA was purified with the PureYield Plasmid Miniprep System (Promega). The recombinant proteins were produced according to the manufacture's protocol. The reaction mixture (12.5 μL) and the buffer (12.5 μL) containing 100 mM Tris-HCl (pH 7.5), 200 mM NaCl, and 0.1% polysorbate 20, and 0.5 nmol of chlorophyll *a* and incubated at 25 °C in the dark for 60 min. After incubation, 200 μL of acetone was added, and the pigments were analyzed with HPLC as previously described[14]. The elution profiles were determined with a fluorescence detector monitoring 680-nm fluorescence with 410-nm excitation (RF-20A; Shimadzu).

**RNA isolation and quantification of RT-PCR analysis.** The total RNA was extracted from various rice tissues with WelPrep total RNA isolation reagent (WELGENE, Republic of Korea), according to the manufacturer's instructions, and treated with RNase-free DNase I (Takara Bio, Shiga, Japan) to prevent genomic DNA contamination. First-strand cDNA was synthesized from 2 μg of total RNA in a 25-μL reaction mixture with using the ImProm II Reverse Transcriptase system kit (Promega, Madison, WI), followed by quantitative real-time PCR (qRT-PCR) analysis to determine gene expression levels (Bio-Rad, CFX96 Touch Real-Time PCR Detection System, USA) using a SYBR premix ExTaq kit (Takara Bio, Shiga, Japan). The gene expressions were normalized using the rice *ubiquitin* gene (LOC_Os06g46770) as an internal control. Changes in expression were calculated via the $\Delta\Delta_{Ct}$ method. Primers for PCR are listed in Supplementary Table 5.

**Generation of transgenic plants.** To make *OsSGR*-overexpression construct, the full-length cDNA sequences were amplified from the flag leaves of IR72 (*indica*) and JN (*japonica*) with primer pair OxF and OxR (Supplementary Table 5). The PCR products were inserted into pGA3426, driven by maize *Ubiquitin* (*Ubi*) promoter[36]. JN was used for producing transgenic plants by *Agrobacterium*-mediated co-cultivation[37]. For the construction of *OsSGR* RNAi vector, pANDA gateway vector[38] was used for cloning the C-terminal regions using primer sets listed in Supplementary Table 5. To generate the Cas9-targeting construct for *OsSGR*, two gene-specific spacer sequences listed in Supplementary Table 5 were cloned into the entry vectors, and then cloned into destination vectors containing the Cas9 expression cassette pH-Ubi-cas9-7, of which the Cas9 coding sequence was codon-optimized for expression in rice and was driven by the maize *Ubi* promoter[39]. These two constructs were transformed into *Oryza sativa* cv. Kasalath[40]. Transgenic plants were grown in the paddy field located at Daegu Gyeongbuk Institute of Science and Technology (Daegu; 35.8° N; 127.6° E) in the year 2017 and 2018.

**Isolation of T-DNA mutants.** Putative *OsSGR* and *OsPRR95* mutant lines in *japonica* cultivar Dongjin were isolated from rice flanking sequence-tag database[41,42]. For genotyping, two gene-specific primers and one T-DNA-specific primer were used. Transcript levels of *OsSGR* or *OsPRR95* were determined by qRT-PCR, using cDNA prepared from 15- and 100-DAG (Days after germination) leaves from WT, *OsSGR-D1* (line 3A-01206), *OsSGR-D2* (line 3A-00334), *OsPRR95-D1* (5A-00143), and *OsPRR95-D2* (3A-13152). Primers for genotyping and qRT-PCR are listed in Supplementary Table 5.

**Tobacco-infiltration assay.** The pCAMBIA1302 vector used for expression of *indica* (IR72)-, *japonica* (JN)- and *ossgr*-type *OsSGR*. cDNAs were amplified by PCR with primers in Supplementary Table 5. After sequencing, PCR products were cloned in pCAMBIA1302 using *Nco*I/*Spe*I restriction sites. The resulting constructs and empty vectors were introduced into *A. tumefaciens* GV3101. Tobacco-infiltration assay was conducted as described previously with minor modifications[43]. Cultures for inoculation were prepared by harvesting *A. tumefaciens* ($OD_{600}$:0.4), and then resuspended the harvested cells in infiltration buffer containing 10 mM MES, pH 5.5; 10 mM $MgSO_4$ and 100 μM acetosyringone. Infiltration with syringe was performed by infiltrating the prepared *Agrobacterium* into 4-weeks-old *Nicotiana benthamiana* leaves. Photos were taken 4 days after infiltration.

**Transient expression assay in rice protoplast.** To generate reporter vectors, the promoters of *OsSGR* (*japonica*: JN and *indica*: IR72) were amplified with the *OsSGR* promoter primer set (Supplementary Table 5). The primers used were as follows: *OsSGR*-pro-F1 and *OsSGR*-pro-R1 for #1 (−2000 to −1 and 5′UTR of *japonica* promoter) and #2 (−2008 to −1 and 5′UTR of *indica* promoter). For luciferase fusion with *OsSGR* promoter, the PCR products were ligated into pGreenII 800-Luc after restriction enzyme treatment. All PCR products were sequenced to confirm the nucleotide sequences. The maize *Ubi* promoter: *β-glucuronidase* (*ZmUbi*: *GUS*) construct was used as an internal control[44].

Protoplasts were isolated from rice root-derived callus suspension (Oc) cells according to the reported method[44], but with minor modifications. Briefly, the Oc suspension solution was collected by centrifugation, and the supernatant was removed. The cells were then incubated in enzyme solution (2% cellulose RS, 1% macerozyme, 0.4 M mannitol, 0.1% MES, pH 5.7, and 0.1% $CaCl_2$) for 4 h with gentle shaking. After washing with equal volume of KMC solution (117 mM KCl, 82 mM $MgCl_2$, and 85 mM $CaCl_2$), harvested protoplasts were resuspended in

MMG solution (0.4 M mannitol, 15 mM $MgCl_2$, and 4 mM MES, pH 5.7) to achieve a density of $3 \times 10^6$ cells $mL^{-1}$, as quantified with a hemocytometer. For transient expression assays, isolated protoplasts were cotransfected with $OsSGR$ promoter reporter constructs using a polyethylene glycol-calcium-mediated method[45]. $ZmUbi: GUS$ was included in each sample as an internal control. Transfected protoplasts were incubated in incubation solution (0.5 M mannitol, 20 mM KCl, 4 mM MES, pH 5.7) for 5 h and then harvested. The harvested protoplasts were resuspended in lysis buffer and used for Luciferase and GUS assays. Luciferase assays were performed using the Luciferase assay system (Promega), and GUS assays were performed by previously described methods[46]. The fluorescence generated by Luciferase and GUS activity was measured by the VICTOR2 1420 multilabel counter (PerkinElmer Life Sciences). In each sample, the measured Luciferase activity was divided by the GUS activity to normalize the data for variation in experimental conditions, and all transient expression experiments were repeated three times with similar results.

**Haplotype analysis and SNP analysis**. 2-kb promoter regions of 105 accessions (Supplementary Data 1) were amplified by PCR using primers listed in Supplementary Table 5. The PCR products were sequenced and used for haplotype analysis. The sequences from accessions were aligned using MEGA6.0[47]. A phylogenetic tree was constructed using the neighbor-joining method in MEGA6.0. Chlorophyll contents were also measured in flag leaves at 6 weeks after heading date using these accessions grown in paddy fields.

**Association of polymorphic sites with chlorophyll contents**. For each polymorphic site in the $OsSGR$ promoter, we divided the accessions into two groups, one with the same DNA base and the other with the different base compared with the reference sequence of $japonica$ Nipponbare (IRGSP-1.0). Next, we compared the chlorophyll contents between the two groups using Student's $t$ test. For multiple testing correction of the measured $T$ value for each site, an empirical null distribution of $T$ values was estimated by performing random permutations of the accessions 1000 times. The adjusted $P$-value was then calculated by applying the two-tailed test to the measured $T$ value for each site using the distribution.

**Phylogeny and selective sweep analysis**. The whole genome sequencing dataset of Huang et al.[27] were reanalyzed to examine the phylogeny of the $OsSGR$ region in an expanded set of wild and domesticated rice sample. Raw sequencing reads were downloaded from the National Center for Biotechnology Information website under bioproject ID numbers PRJEB2052, PRJEB2578, and PRJEB2829, for a total of 1477 rice samples. Raw sequencing reads were trimmed using trimmomatic ver. 0.36[48], and realigned to the reference Nipponbare genome (MSU7/IRGSP-1.0) using BWA-MEM ver. 0.7.15[49]. Because the sequencing depth was very shallow (~1 to 2×) for each sample, we used genotype probability and likelihood for all downstream analysis to incorporate the uncertainty associated with the low coverage dataset.

The programs ANGSD ver. 0.913[50] and ngsTools[51] were used while using values from Choi and Purugganan[52] during parameter specifications. Phylogenetic tree of the $OsSGR$ region was reconstructed by estimating the genotype probabilities of 10-kb upstream and downstream of the $OsSGR$ gene. The genotype probabilities were then used to estimate a pairwise genetic distance matrix[53], which were then used to build a neighbor-joining tree with FastME ver. 2.1.5[54].

Evidence of a domestication-related selective sweep was examined by estimating the ratio of wild to domesticated rice polymorphism levels ($\pi_W/\pi_D$). Because domestication-mediated selection would reduce the level of polymorphism in only the domesticated but not the wild rice, $\pi_W/\pi_D$ values will be elevated for genomic regions that have undergone a domestication-related selective sweep[27]. ANGSD was used to estimate the level of polymorphism in 20 kb non-overlapping windows for each wild and domesticated rice subpopulation. The level of polymorphism for each rice subpopulation ($aus$, $indica$, and $japonica$) was compared to the Or-II group of wild rice, which was previously determined to be the most genetically diverged wild rice group from all domesticated rice subpopulations[27,52]. A window was assumed to be significant if the $\pi_W/\pi_D$ value was greater than the 1% empirical distribution of $\pi_W/\pi_D$ values.

**Measurement of Fv/Fm ratio**. The $Fv/Fm$ ratio was measured using a plant efficiency analyzer (PEA) (Hansatech, Norfolk, UK) following the manufacturer's instructions. After the dark adaptation of the middle part of each flag leaves for 20 min, the $Fv/Fm$ ratio was recorded in the paddy field. More than four experimental replicates were conducted.

**Measurement of net $CO_2$ assimilation rate**. Plants were transferred from rice paddy field to glasshouse after heading and adapted in the glasshouse for 2 weeks to examine $CO_2$ assimilation rate. $CO_2$ assimilation rate was measured with portable gas exchange system LI-6850 (LI-COR Inc., Lincoln, NE, USA) following saturating irradiance of 1500 µmol $m^{-2}$ $s^{-1}$, $CO_2$ concentration of 400 ppm, relative humidity of 60%, fan speed of 10,000 rpm, and flow rate of 600 µmol $s^{-1}$ between 11 am to 2 pm.

**Measurement of relative growth rate**. Three plants of each genotypes were carefully sampled and washed to remove the roots and soil. Total dry weight of each plant was measured after drying the plant in an oven at 70 °C to a constant weight. RGR is calculated according to the following formula[28,29]: RGR = (lnW2 − lnW1)/(t2 − t1) where: ln = natural log, W1 = dry weight of plant at time t1 (in grams), W2 = dry weight of plant at time $t2$, $t1$ = time one (in days), $t2$ = time 2, respectively and is expressed as g $g^{-1}$ $day^{-1}$.

## Data availability

All data and analysis needed to understand and evaluates the conclusions in the paper are present in the paper or Supplementary Materials. The source data underlying Figs. 1d–g, i, 2f–h, 3a, c and 4g–m and Supplementary Figs. 1, 2a, b, 3, 4b, 5b–d, g, 5i–m, 6c, d, f, h–m, 7b, c, 8b, d, e, g, i, 10b, e–i, 11a, 16c–g and 17a–j are provided as a Source Data file.

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

## Acknowledgements

We thank Dr. Gi-Gyeong Park and Kyungsook An for taking care of the transgenic rice plants; Nam-Chon Paek for sharing *ossgr1* seeds; Makoto Kusaba for *ossgr2* seeds. We thank Life Science Editors for editorial assistance. This research was supported by the Institute for Basic Science (IBS-R013-D1) from the Ministry of Science, ICT & Future Planning, and by the Research Program for Agricultural Science and Technology Development (Project No. PJ01099902) Rural Development Administration.

## Author contributions

H.G.N. and S.L. conceived and supervised the project. H.G.N., S.L., D.S., and J.H.L. designed the experiments. D.S., T.H.K., J.Y. L., J.H.C., Y.C.S., and J.H.L. developed the NILs. D.S., S.L., J.P., and T.H.K. conducted the QTL analysis and field experiments. G.A., L.H.C., and J.L. provided the T-DNA mutants and generated the transgenic plants. D.W. L., J.L., J.S.J., and D.H.K. performed the protoplast experiments. W.L., J.H.P., D.H., J.Y.C., S.L., and M.D.P. performed the bioinformatics analysis. H.I., A.T., and J.W.L. conducted the biochemical assay and in planta infiltration assay of rice OsSGR. D.S., S.L., T.H.K., J.H.L., and H.G.N. analyzed the data and wrote the paper. All the authors discussed the results and contributed to the paper.

## Competing interests

The authors declare no competing interests.
