## [Peer Review File · Nature Communications]

Reviewers' comments:

Reviewer #1 (Remarks to the Author):

Starting with an interest on the difference in the speed of leaf senescence after heading between indica rice (early senescence in general) and japonica rice (late senescence in general), authors cloned a causal gene for the different senescence pattern by mapping-based approach. The gene encodes a reported chlorophyll-degrading protein OsSGR that has been functionally characterized. After confirming the genetic effect of OsSGR gene on leaf and panicle senescence, this work further explored the natural variation of this gene and found the variations in the promoter, rather than coding sequence, determine the difference in senescence between indica and japonica, and revealed that the indica and japonica alleles are originated from nivara and rufipogon rice, respectively, which agrees with evolution rout of the two subspecies of cultivated rice. The work also demonstrated the potential value of this gene in increasing yield by introducing the japonica allele for late senescence into india cultivars.

Although the OsSGR has been reported for its function in delaying senescence and post-flowering leaf stay-green has also been recognized for its effect in increasing grain filling and yield, this work provided solid genetic evidence and explored natural variation of the gene together with the origination of the indica and japonica allele, and proposed a evolutionary mechanism of the senescence and death for the r-selection life history trait. Therefore, this work provided very limited new knowledge on the function of the OsSGR protein in senescence but some useful information on the variation, evolutionary history and potential breeding value of this gene. However, for the potential value in breeding (high yield), authors may need consider how to balance the degree of stay-green traits because over stay-green, or greed for green after heading is not desirable and even bad for rice production. In addition, according to my expertise, variations in leaf and panicle senescence also exist within india or japonica rice, and breeders have selected optimal life span including leaf senescence in a specific rice-growing region. Wise strategy is needed to explore this gene in breeding.

In general, the data in this work is of high quality. However, I noticed the transgenic rice overexpressing the OsSGR genes, activation lines and the KO mutant were not shown for the grain-filling rate and yield. Although the NILs in indica background with japonica allele showed significant increased in grain-filling rate and yield, it cannot be excluded that the effect is contributed by other unknown genes since no data to show how 'near' the NILs are. It could be more conclusive if grain-filling rate and yield data is collected for the OE and mutant.

Reviewer #2 (Remarks to the Author):

Leaf senescence is the final stage of leaf development and the enlargement of this period is related to grain yield in rice. Between indica and japonica, two species, there is difference in this period and the indica rice has a rapid life cycle (early senescence). In this report, Shin D and his colleague indicated that natural variation in OsSGR promoter region might determine the lifespan variations between two species and the modify in this region could enhance grain yield.

Unfortunately, I regret to inform that this manuscript cannot be considered for publication in Nature Comm. because it does not meet the requirements of the journal at least in this version.

Major point

1. First, ten or more genes have already been identified as the genes that cause stay-green (e.g., Leng et al., *Int. J. Mol. Sci.* 2017). Therefore, it is difficult to think that the lifespan variation between japonica and indica species can be determined by only one gene examined in this paper. The authors should clarify if there is a similar variation at least in several major genes.

2. It also relates to the relationship between the lifespan and the yield, which is another main theme of this paper. The idea that flag leaves, which are the main source organs of rice, could maintain photosynthetic ability for a long period and continue substance production leads to an

increase in yield. Meanwhile, it was reported that OsSGR (senescence-inducible chloroplast stay-green protein I), the gene examined in this paper maintained the green color of leaves but did not affect photosynthetic ability and yield (Jiang H et al., Plant J. 2007). Authors should analyze photosynthetic ability of flag leaves or a canopy through growth stage and add the data of growth analysis (NAR) to clarify factors of yield characteristics.

Minor point

1. The authors should add the information about stay-green of rice or Arabidopsis into first paragraph and explain the relation between lifespan and senescence.
2. pp.4 line 88, please add accession number of OsSGR.
3. pp.8, Authors should discuss about the reason why the specific promoter region that causes short lifespan has been selected in indica but not in japonica through the domestication.

Reviewer #3 (Remarks to the Author):

In this manuscript it has been convincingly demonstrated that yield of an important crop plant, i.e. rice, can be enhanced by delaying the senescence process. This is a result of high importance with regard to the increasing demand in food supply.

The study has been performed with two agronomically important rice varieties having contrasting senescence processes. The molecular reason underlying the fast senescence feature of indica rice in comparison to japonica rice has been elucidated by QTL analysis and a broad repertoire of further sophisticated molecular analyses. The study shows that variations in the promoter of the so-called STAYGREEN gene encoding an enzyme of chlorophyll catabolism are responsible for high expression of the gene and the fast senescence phenotype of the rice variety indica. By investigating the STAYGREEN sequences of wild rice species the researchers authors could track the evolution of the sequences responsible for the fast senescence of indica rice and demonstrate that introgression of STAYGREEN alleles from japonica into indica cultivars increased yield. The results here presented have been obtained by a multitude of long-standing elaborate studies including field trials. A considerable part of the data is contained in the supplement. All data is presented sufficiently detailed and transparent to allow for an understanding of this very impressive work. Appropriate statistical tests have been done and are well documented. The conclusions are robust, valid and reliable. Indeed an increase in yield of about 10% is quite substantial. If the space would allow it, a comment on the different staygreen characters might be appropriate. Not in all conditions a staygreen character is advantageous, as detailed in the review of Gregersen et al. (2013, PMB).

Specific comments:

Line 25-26 - The introduction about the importance of senescence for crop yield should reference the highly cited paper on senescence and crop productivity of Gregersen et al. (2013, Plant Molecular Biology).

Line 43: r-selection has to be explained to make the paper accessible for non-experts

Line 60 and later in the text: The numbering of leaves should be changed. The flag leaves is certainly not the first leaf, but actually the last leaf. Accordingly, the "second" leaf is not the second leaf, but the first leaf below the flag leaf. The first leaf is the primary foliage leaf! The flag leaf is the leaf with the highest number, which depends on the species and variety.

Line 88: For non-expert readers it might be misunderstanding that a gene called STAYGREEN accelerates senescence. The authors should clearly indicate that the gene has been detected to be impaired in a staygreen mutant and hence has been named accordingly.

Last but not least the style of writing should be changed. Please avoid the use of the first person and write the complete manuscript in the third person. The first person is only appropriate for beliefs and convictions and is usually used in autobiographies. The description of scientific results rather requires impersonal reporting using third person pronouns instead of "we", because the results hopefully exist without the reflections of the writing persons.

Reviewer #4 (Remarks to the Author):

The manuscript "Natural variations at the Stay Green gene promoter control lifespan and yield in rice cultivars" describes the identification of OsSGR as the gene responsible for the difference in lifespan between the two major rice subspecies, japonica and indica. Analysis on sequences of OsSGR in different rice accessions suggested that sequence variations in the promoter region are critical in the evolution of the indica subspecies as a rapid cycling rice in which higher expression of the OsSGR gene triggers earlier senescence of leaves and panicles. Results from this study could potentially bring about a better understanding of rice evolution in light of plant senescence and the authors have proposed potential application of this locus in rice breeding to increase yield.

One critical question needs to be addressed, however, is the discrepancy between the increase in yield in rice plants with reduced OsSGR expression in this study, and the results from earlier studies indicating that *sgr* is a nonfunctional stay-green locus (Park et al., 2007, *Plant Cell*; Sato et al., 2007, *PNAS*). For nonfunctional stay-green mutants, the plants retain high chlorophyll contents in senescent leaves but their photosynthetic competence decreases normally during senescence. This type of mutations is often associated with loss of function in genes involved in chlorophyll degradation, which is believed to be at relatively downstream part of the senescence program. The fact that OsSGR encodes for a chlorophyll-degrading Mg²⁺-dechelatase, is consistent with *sgr* being a nonfunctional stay-green mutant. In this study, however, the authors showed that when OsSGR expression was enhanced by introgression of the japonica OsSGR allele into indica type cultivars, the plants exhibited delayed senescence and increased yield.

I would suggest the author to address this question by measuring photosynthesis competence and examining senescence marker gene expression in the mutants, transgenic plants, and introgression lines to show that indeed enhanced expression of OsSGR is able to delay the senescence program – not just chlorophyll degradation. If that's the case, the authors also need to explain why in this case, the whole senescence process can be delayed by simply blocking a step in chlorophyll degradation.

Reviewers' comments:

Reviewer #1 (Remarks to the Author):

Starting with an interest on the difference in the speed of leaf senescence after heading between indica rice (early senescence in general) and japonica rice (late senescence in general), authors cloned a causal gene for the different senescence pattern by mapping-based approach. The gene encodes a reported chlorophyll-degrading protein OsSGR that has been functionally characterized. After confirming the genetic effect of OsSGR gene on leaf and panicle senescence, this work further explored the natural variation of this gene and found the variations in the promoter, rather than coding sequence, determine the difference in senescence between indica and japonica, and revealed that the indica and japonica alleles are originated from nivara and rufipogon rice, respectively, which agrees with evolution rout of the two subspecies of cultivated rice. The work also demonstrated the potential value of this gene in increasing yield by introducing the japonica allele for late senescence into india cultivars. Although the OsSGR has been reported for its function in delaying senescence and post-flowering leaf stay-green has also been recognized for its effect in increasing grain filling and yield, this work provided solid genetic evidence and explored natural variation of the gene together with the origination of the indica and japonica allele, and proposed an evolutionary mechanism of the senescence and death for the r-selection life history trait. Therefore, this work provided very limited new knowledge on the function of the OsSGR protein in senescence but some useful information on the variation, evolutionary history and potential breeding value of this gene. However, for the potential value in breeding (high yield), authors may need consider how to balance the degree of stay-green traits because over stay-green, or greed for green after heading is not desirable and even bad for rice production.

-Response: We agree with reviewer's concern. Stay-green refers to the heritable delay of leaf senescence, which has been identified as an important component for the improving crop yield through extending of photosynthesis period in various crop species. For example, progressive increases in grain yield of maize were positively correlated with stay-green traits. However, most of previous results for stay-green characters in rice were studied by loss-of-function mutants and/or overexpression lines of specific genes, leading to over stay-green together with undesirable crop productivity, as the reviewer pointed out. For example, *ossgr* mutants exhibiting delayed leaf senescence did not show an advantage in grain yield.

The key point in our report is that the *japonica* *OsSGR* allele does not lead to over-stay green but shows difference in its induction kinetics and induction level compared to the *indica* allele. Thus, the *japonica* allele exhibits a capacity to balance the degree of stay-green traits when introduced into *indica*

varieties, as we have shown in this report that NILs-*OsSGR* in the *indica* varieties harboring a *japonica* *OsSGR* allele leads to delay of senescence in a *japonica* fashion with balanced stay-green in *indica*, not like *stay-green* knock out mutants. As a results, NILs showed the improved productivity along with the enhanced grain filling, which was useful for breeding purpose. We added this argument in Discussion.

In addition, according to my expertise, variations in leaf and panicle senescence also exist within *indica* or *japonica* rice, and breeders have selected optimal life span including leaf senescence in a specific rice-growing region. Wise strategy is needed to explore this gene in breeding.

-Response: We thank for the suggestion and we agree. As the reviewer pointed out, there are a lot of *indica* and *japonica* varieties showing different lifespan including heading date. Lifespans of rice varieties are also dependent on their cultivated environments such as latitude and growing season. We understand that our data is based on the Korean rice field and on the elite cultivars bred in Korea and that the traits may be somewhat different in different areas and in different cultivars. It would be necessary to consider the varieties and the local environments in actual use of this allele in breeding in other areas.

In general, the data in this work is of high quality. However, I noticed the transgenic rice overexpressing the *OsSGR* genes, activation lines and the KO mutant were not shown for the grain-filling rate and yield. Although the NILs in *indica* background with *japonica* allele showed significant increase in grain-filling rate and yield, it cannot be excluded that the effect is contributed by other unknown genes since no data to show how 'near' the NILs are. It could be more conclusive if grain-filling rate and yield data is collected for the OE and mutant.

-Response: We thank for this comment. In this revised manuscript, we included the agronomic traits of activation lines, overexpressing transgenic lines and *ossgr* mutants along with their WT (Supplementary Fig. 5j-n, Supplementary Fig. 6i-m, Supplementary Fig. 10e-i), as the reviewer asked. OX and activation lines, displayed the decreased grain filling rate, leading to less total grain yield per plants, as compared with WT. Two *ossgr* mutants did not show the yield advantage (Supplementary Fig. 6i-m).

Reviewer #2 (Remarks to the Author):

Leaf senescence is the final stage of leaf development and the enlargement of this period is related to grain yield in rice. Between *indica* and *japonica*, two species, there is difference in this period and the *indica* rice has a rapid life cycle (early senescence). In this report, Shin D and his colleague indicated that natural variation in *OsSGR* promoter region might determine the lifespan variations between two species

and the modify in this region could enhance grain yield. Unfortunately, I regret to inform that this manuscript cannot be considered for publication in Nature Comm. because it does not meet the requirements of the journal at least in this version.

Major point

1. First, ten or more genes have already been identified as the genes that cause stay-green (e.g., Leng et al., Int. J. Mol. Sci. 2017). Therefore, it is difficult to think that the lifespan variation between japonica and indica species can be determined by only one gene examined in this paper. The authors should clarify if there is a similar variation at least in several major genes.

-Response: We understand the reviewer's concern. As reviewer has pointed, leaf senescence is controlled by a complex set of genes with sophisticated processes. We agree with the reviewer that there will be other genes that lead to the different senescence between *indica* and *japonica* varieties. We also described in this report that there are several QTL loci that contribute to the different senescence between *indica* and *japonica* varieties (Fig. 2a, b). In this report, we focused on one of those loci that significantly contribute to the different senescence phenotype between *indica* and *japonica* varieties. We added the point of the reviewer in Introduction with the reference (page 3, line 63-66). We like to note that we are trying to narrow down for another leaf senescence QTLs as a separate project.

2. It also relates to the relationship between the lifespan and the yield, which is another main theme of this paper. The idea that flag leaves, which are the main source organs of rice, could maintain photosynthetic ability for a long period and continue substance production leads to an increase in yield. Meanwhile, it was reported that OsSGR (senescence-inducible chloroplast stay-green protein I), the gene examined in this paper maintained the green color of leaves but did not affect photosynthetic ability and yield (Jiang H et al., Plant J. 2007). Authors should analyze photosynthetic ability of flag leaves or a canopy through growth stage and add the data of growth analysis (NAR) to clarify factors of yield characteristics.

-Response: We thank for the reviewer's suggestion. In the revised manuscript, we added the data, as the reviewer requested. Following the reviewer's comment, we analyzed the photosynthetic ability of flag leaves of four NILs grown in field to evaluate the effect on grain yield. Three NILs of the *indica* background harboring *japonica* *OsSGR* allele displayed extended the photosynthetic capability with higher chlorophyll contents (Fig. 4g, i, and k, and Supplementary Fig. 17b), leading to the improved grain-filling rate and grain yield. Therefore, we inferred that *japonica* *OsSGR* allele in *indica* background

could increase the grain productivity by maintaining a higher chlorophyll content and photosynthetic capability of flag leave during grain filling period. On the contrary, NILs of the *japonica* background harboring *indica* *OsSGR* allele displayed decreased the photosynthetic capability with less chlorophyll contents, resulting in poor grain filling rate and productivity, as compared with parental *japonica* (Fig. 4g, i, k).

Minor point

1. The authors should add the information about stay-green of rice or Arabidopsis into first paragraph and explain the relation between lifespan and senescence.

-Response: As suggested by reviewer, we added information about stay-green in the text (pp 3, line 55-63).

2. pp.4 line 88, please add accession number of *OsSGR*.

-Response: As suggested by the reviewer, we included the accession number of three candidate genes (page 5, line 119-120).

3. pp.8, Authors should discuss about the reason why the specific promoter region that causes short lifespan has been selected in *indica* but not in *japonica* through the domestication.

-Response: We thank for this comment. According to the reviewer's comment, we added the following point in the Discussion part (page12, line 259-277). The following is our idea. Optimal rice lifespan of varieties has been considered and selected for maximum productivity and/or economic income according to their cultivation climate and cropping system. In the single cropping region including most of *japonica* cultivation area, grain filling rate is important agricultural trait for maximum crop production. Therefore, the lower transcript levels of *OsSGR* in *japonica* could extend photosynthesis capacity during grain filling period for improved productivity. On the contrary, most of *indica* cultivation region has double or triple cropping system. Therefore, cultivation period with maximum production should be considered. For this reason, promoter of *OsSGR* in *indica* was naturally selected for the faster and higher expression level of *OsSGR*. However, natural selection on coding region of *OsSGR* was not occurred during domestication because *ossgr* mutants exhibiting impaired chlorophyll breakdown did not affect productivity. We described our idea on why promoter region was selected for controlling of life span during domestication in the manuscript (page8, line 181-194)

Reviewer #3 (Remarks to the Author):

In this manuscript it has been convincingly demonstrated that yield of an important crop plant, i.e. rice, can be enhanced by delaying the senescence process. This is a result of high importance with regard to the increasing demand in food supply. The study has been performed with two agronomically important rice varieties having contrasting senescence processes. The molecular reason underlying the fast senescence feature of indica rice in comparison to japonica rice has been elucidated by QTL analysis and a broad repertoire of further sophisticated molecular analyses. The study shows that variations in the promoter of the so-called STAYGREEN gene encoding an enzyme of chlorophyll catabolism are responsible for high expression of the gene and the fast senescence phenotype of the rice variety indica. By investigating the STAYGREEN sequences of wild rice species the researchers authors could track the evolution of the sequences responsible for the fast senescence of indica rice and demonstrate that introgression of STAYGREEN alleles from japonica into indica cultivars increased yield. The results here presented have been obtained by a multitude of long-standing elaborate studies including field trials. A considerable part of the data is contained in the supplement. All data is presented sufficiently detailed and transparent to allow for an understanding of this very impressive work. Appropriate statistical tests have been done and are well documented. The conclusions are robust, valid and reliable. Indeed an increase in yield of about 10% is quite substantial. If the space would allow it, a comment on the different staygreen characters might be appropriate. Not in all conditions a staygreen character is advantageous, as detailed in the review of Gregersen et al. (2013, PMB).

We thank for the positive evaluation. It is well established that there are different stay-green characters. We added this point in Introduction.

Specific comments:

Line 25-26 - The introduction about the importance of senescence for crop yield should reference the highly cited paper on senescence and crop productivity of Gregersen et al. (2013, Plant Molecular Biology).

-Response: As reviewer suggested, we included this review article as a reference in the text (page 3, line 52-61).

Line 43: r-selection has to be explained to make the paper accessible for non-experts

-Response: We thanks for reviewer's comment. We revise the text as reviewer suggested (page 2, 3, line 43-47)

Line 60 and later in the text: The numbering of leaves should be changed. The flag leaves is certainly not

the first leaf, but actually the last leaf. Accordingly, the “second” leaf is not the second leaf, but the first leaf below the flag leaf. The first leaf is the primary foliage leaf! The flag leaf is the leaf with the highest number, which depends on the species and variety.

-Response: We thank for the comments. According to the reviewer’s comment, we revised the numbering of leaves through the manuscript; first leaf into flag leaf and second leaf into second uppermost leaf.

Line 88: For non-expert readers it might be misunderstanding that a gene called STAYGREEN accelerates senescence. The authors should clearly indicate that the gene has been detected to be impaired in a staygreen mutant and hence has been named accordingly.

-Response: We totally agree with the reviewer’s concern. We included the meaning of the ‘stay-green’ term, which used for mutations in chlorophyll breakdown (page 3, line 58-61).

Last but not least the style of writing should be changed. Please avoid the use of the first person and write the complete manuscript in the third person. The first person is only appropriate for beliefs and convictions and is usually used in autobiographies. The description of scientific results rather requires impersonal reporting using third person pronouns instead of “we”, because the results hopefully exist without the reflections of the writing persons.

-Response: We agree with the reviewer’s concern. We revised the text through the manuscript according to the reviewer’s suggestion.

Reviewer #4 (Remarks to the Author):

The manuscript “Natural variations at the Stay Green gene promoter control lifespan and yield in rice cultivars” describes the identification of OsSGR as the gene responsible for the difference in lifespan between the two major rice subspecies, japonica and indica. Analysis on sequences of OsSGR in different rice accessions suggested that sequence variations in the promoter region are critical in the evolution of the indica subspecies as a rapid cycling rice in which higher expression of the OsSGR gene triggers earlier senescence of leaves and panicles. Results from this study could potentially bring about a better understanding of rice evolution in light of plant senescence and the authors have proposed potential application of this locus in rice breeding to increase yield.

One critical question needs to be addressed, however, is the discrepancy between the increase in yield in rice plants with reduced OsSGR expression in this study, and the results from earlier studies indicating that sgr is a nonfunctional stay-green locus (Park et al., 2007, Plant Cell; Sato et al., 2007, PNAS). For

nonfunctional stay-green mutants, the plants retain high chlorophyll contents in senescent leaves but their photosynthetic competence decreases normally during senescence. This type of mutations is often associated with loss of function in genes involved in chlorophyll degradation, which is believed to be at relatively downstream part of the senescence program. The fact that OsSGR encodes for a chlorophyll-degrading Mg^{2+} -dechelatase, is consistent with *sgr* being a nonfunctional stay-green mutant. In this study, however, the authors showed that when OsSGR expression was enhanced by introgression of the japonica OsSGR allele into indica type cultivars, the plants exhibited delayed senescence and increased yield. I would suggest the author to address this question by measuring photosynthesis competence and examining senescence marker gene expression in the mutants, transgenic plants, and introgression lines to show that indeed enhanced expression of OsSGR is able to delay the senescence program – not just chlorophyll degradation. If that's the case, the authors also need to explain why in this case, the whole senescence process can be delayed by simply blocking a step in chlorophyll degradation.

-Response: We understand the reviewer's concern. As a response to this and another reviewer, we now included the photosynthetic competence of flag leaves from four NILs grown in field. As described in detail below, our results clearly show that timing and level of *OsSGR* expression can control the whole senescence program in rice during grain filling period by regulating the photosynthetic capability not just the chlorophyll content.

The reason is as follows. Delay of chlorophyll loss may not be directly related to functional delay of senescence as well known (Park et al., 2007). However, it is clear and well known that late degradation of chlorophyll leads to later functional senescence, as reported in many cases (Yoo et al., 2007, Mol Cells.; Liang et al., 2014, PNAS). Thus, as we reported here, *indica* varieties show earlier chlorophyll loss and accordingly earlier senescence through acquisition of a specific promoter sequence of the *OsSGR* gene during their evolution, which causes early and high-level induction of the *OsSGR* gene and lead to a short lifespan and fast cycling. We have replaced this early chlorophyll loss allele of *indica* varieties with a progenitor allele present in *japonica* varieties that does not cause early chlorophyll loss and early senescence. The result was that “not early chlorophyll loss” due to the introduction of the *japonica* allele led to longer photosynthetic period.

The word ‘Stay green’ for the *OsSGR* gene can be easily misunderstood, as the knock out mutants shows ‘stay green’ phenotype (page 3, line 59-61), but the actual function of the *OsSGR* gene is chlorophyll degradation and the concomitant leaf senescence. We added this point in the text (page 5, 6, line 120-122).

Below, we describe the photosynthetic capacity data in detail. According to our analysis, three *indica* NILs having *japonica* *OsSGR* allele maintained their photosynthetic capability longer than their parental varieties (Fig. 4k and Supplementary Fig. 17b), and *japonica* NIL harboring *indica* *OsSGR* allele displayed decreased photosynthetic capability (Fig. 4k). As a senescence marker gene, we analyzed the expression of *OsNAP* (LOC_Os03g21060) in rice plants. Activation lines showing early senescence accumulated higher levels of *OsNAP* transcripts in flag leaves (Supplementary Fig. 5d), delayed leaf senescence mutants including *ossgr*, CRISPR/Cas9-genome editing and RNAi-suppressed lines displayed the decreased expression of *OsNAP*, confirming their leaf senescence phenotype (Supplementary Fig. 6d, Supplementary Fig. 7c and Supplementary Fig. 8e). The expression of *OsNAP* in *indica* (IR72)-NIL harboring *japonica* *OsSGR* allele was lower, as compared to IR72, and its expression was higher in the *japonica*-NIL having *indica* *OsSGR* allele compared to parental *japonica* variety (Supplementary Fig. 17a).

Reviewers' comments:

Reviewer #2 (Remarks to the Author):

In this report, Shin D and his colleague indicated that natural variation in OsSGR promoter region might determine the lifespan variations between two species and the modify in this region could enhance grain yield. In previous review, I pointed out that authors should analyze photosynthetic ability of flag leaves or a canopy through growth stage and add the data of growth analysis (NAR) to clarify factors of yield characteristics. Because it is most important and valuable point of this report to clarify which the lower decrease in chl. content by their allele can maintain the photosynthetic ability and then increase in yield or not. However, in present version, they added the Fv/Fm (chlorophyll fluorescence) data in Figure 4k and Supplementary Fig. 17b. Their data could not indicate higher photosynthetic ability. For example, between IR72 and NIL the larger difference in chl. content did not reflect photosynthetic ability after 7 weeks after heading (Fig. 4k). Additionally, Fv/Fm value does not necessary match the photosynthetic ability (Harley et al. Plant Physiol. 1992). Unfortunately, I regret to inform that this manuscript cannot be considered for publication without adequate data and thorough discussion.

Minor point

1. Authors should add the statistical analysis in suppl. Fig. 3 and Table 4.

Reviewer #3 (Remarks to the Author):

I accept the replies to my comments on the first version of this manuscript. However, I'm disappointed that the style of writing has not been changed throughout the text.

I hence repeat:

Please avoid the use of the first person (I, we) and write the complete manuscript in the third person (e.g. it has been shown...). The first person is only appropriate for beliefs and convictions and is usually used in autobiographies. The description of scientific results rather requires impersonal reporting using third person pronouns instead of "we", because the results hopefully exist without the reflections of the writing persons.

In the abstract in lines 23, 25, 31 and later in line 67 etc. the personal style has been used.

Reviewer #4 (Remarks to the Author):

The authors have nicely addressed my concern about OsSGR being functional in regulating the senescence program (not just chlorophyll degradation) by showing that the indica NILs having japonica OsSGR allele maintained their photosynthetic capability longer than their parental varieties, and japonica NIL harboring indica OsSGR allele displayed decreased photosynthetic capability. They also analyzed the expression of the senescence marker gene OsNAP in rice plants. Activation lines showing early senescence accumulated higher levels of OsNAP transcripts in flag leaves while delayed leaf senescence mutants including *ossgr*, CRISPR/Cas9-genome editing and RNAi-suppressed lines displayed the decreased expression of OsNAP. The expression of OsNAP in indica (IR72)-NIL harboring japonica OsSGR allele was lower, as compared to IR72, and its expression was higher in the japonica-NIL having indica OsSGR allele compared to parental japonica variety.

My second concern, about the discrepancy between OsSGR being a functional senescence regulator in this study and the results from earlier studies indicating that *sgr* is a nonfunctional stay-green locus (Park et al., 2007, Plant Cell; Sato et al., 2007, PNAS), is however not addressed. In Line 252-258, the authors stated that "The japonica OsSGR allele does not lead to an over stay-

green but shows difference in its induction kinetics and induction level compared to the indica allele (Fig 3a). The japonica allele appears to exhibit a capacity to balance the degree of stay-green trait, photosynthetic competence, nutrient remobilization, when introduced into indica varieties, to lead to increased grain filling and productivity during the senescence period. Thus, NILs-OsSGR in the indica background harboring a japonica OsSGR allele exhibit the balanced stay-green with a delay of senescence, but unlike the stay-green knock out mutants that exhibit unbalanced and non-functional stay-green phenotype". This statement is not convincing to me because based on the new data presented in Supplementary Figure 6 and Supplementary Figure 7, knockout mutants *ossgr* generated via CRISPR/Cas9 genome editing did show significant decrease in expression of the senescence marker gene *OsNAP*, implying that *OsSGR* is a regulator of senescence and loss-of-function in *OsSGR* did cause delayed leaf senescence and theoretically increased photosynthetic capability and yield in this study. I suggest the authors to address this question seriously because *sgr* being a nonfunctional stay-green locus has been reported in several articles in multiple plant species.

Reviewers' comments:

Reviewer #2 (Remarks to the Author):

In this report, Shin D and his colleague indicated that natural variation in *OsSGR* promoter region might determine the lifespan variations between two species and the modify in this region could enhance grain yield. In previous review, I pointed out that authors should analyze photosynthetic ability of flag leaves or a canopy through growth stage and add the data of growth analysis (NAR) to clarify factors of yield characteristics. Because it is most important and valuable point of this report to clarify which the lower decrease in chl. content by their allele can maintain the photosynthetic ability and then increase in yield or not. However, in present version, they added the Fv/Fm (chlorophyll fluorescence) data in Figure 4k and Supplementary Fig. 17b. Their data could not indicate higher photosynthetic ability. For example, between IR72 and NIL the larger difference in chl. content did not reflect photosynthetic ability after 7 weeks after heading (Fig. 4k). Additionally, Fv/Fm value does not necessary match the photosynthetic ability (Harley et al. Plant Physiol. 1992). Unfortunately, I regret to inform that this manuscript cannot be considered for publication without adequate data and thorough discussion.

-Response: We thank Reviewer #2 for these critical suggestions. In the revised manuscript, we added the requested data. Following the reviewer's comment, we measured the net CO₂ assimilation rate of flag leaves to evaluate the effect of *OsSGR* on photosynthetic ability in IR72-NIL and JN-NIL grown in the field (Fig. 4k, Supplementary Fig. 17h). The net CO₂ assimilation rate decreased continuously after flowering in the flag leaves (Fig. 4k). However, the extent of the decrease in the net CO₂ assimilation rate in IR72-NIL was significantly less than that of IR72 (Fig. 4k). At 7 weeks after heading, the net CO₂ assimilation rate and Fv/Fm ratio of IR72-NIL were still higher than those of IR72, which is consistent with the higher levels of chlorophyll in the flag leaves (Fig. 4g, i, k and Supplementary Fig. 17b). IR72-NIL of the *indica* background harboring the *japonica* *OsSGR* allele exhibited extended photosynthesis with higher chlorophyll contents (Fig. 4g, i, and k, and Supplementary Fig. 17b, c), resulting in enhanced grain filling rate and therefore grain yield (Fig. 4l, m and Supplementary Table. 4b). Therefore, *indica* NILs harboring *japonica* *OsSGR* allele displayed extended photosynthetic abilities and therefore increased productivity, similar to functional stay-green genotype. However, *OsSGR* is a gene involved in chlorophyll degradation. The *japonica* alleles of *OsSGR* are progenitors of the *indica* alleles. The *indica* alleles lose chlorophyll earlier than *japonica* alleles due to natural variations in the promoter that activate the expression *OsSGR* earlier during the senescence period in our field condition in a temperate zone. Thus, the *indica* allele exhibits earlier senescence and accelerates the loss of photosynthetic capacity. In contrast, the *japonica* allele leads to

typical slower senescence pattern observed for the *japonica* cultivars grown in temperate zones. Thus, when *japonica* alleles are introduced into *indica* cultivars, the NILs undergo the *japonica* type change of photosynthetic capacity during senescence period, which is slower than the parental *indica* cultivars.

Conversely, JN-NIL of *japonica* harboring the *indica* *OsSGR* allele displayed shortened photosynthetic activity with decreased chlorophyll levels, leading to the decreased grain filling rate and yield, as compared with parental *japonica* (Fig. 4l, m, Supplementary Table 4b and Supplementary Fig. 17e, h, i).

Along with analyzing the photosynthetic ability of NILs grown in the field, we also measured the relative growth rate (RGR) of NILs, as requested. The relative growth rates (RGR) quantifies plant growth speed and is considered to be a reliable standard for estimating plant productivity^{1,2}. It is calculated as the dry mass increment per aboveground biomass at a given time point. RGRs of IR72-NIL are shown in Supplementary Fig. 17d. The RGR of all the rice plants we examined declined continuously during the grain filling stage. However, IR72-NIL harboring *japonica* *OsSGR* allele maintained higher RGR, especially between 5 -7 weeks after flowering, than the parental IR72 (Supplementary Fig. 17d), indicating the contribution of the *japonica* *OsSGR* allele in higher biomass productivity. On the contrary, JN-NIL harboring *indica* *OsSGR* allele showed a relatively faster decline in RGR than its parental cultivar, Junam (Supplementary Fig. 17j).

To further test the yield potential of NILs, large-scale field trials of NILs were performed in the paddy field in 2019. The figure below shows NILs grown in the field at late grain filling stage (Response Figure Fig.1 a-c). When *indica*-NILs harboring *japonica* *OsSGR* allele and their parental *indica* varieties were cultivated in the field, parental varieties were lodged by early senescence, whereas *indica*-NILs remained standing in the field due to the delayed senescence induced by *japonica* *OsSGR* allele (Response Figure 1a-c). Lodging-resistance is important for rice breeding because this is a favorable trait for improving crop productivity³. Furthermore, we experienced typhoons three times during the grain-filling period in this year. JN-NIL harboring *indica* *OsSGR* allele exhibited earlier senescence than its parental *japonica* (Response Figure 1d).

Response Figure 1 Senescence phenotypes of NILs at late grain filling stage grown in the field. Pictures of the representative plant senescence phenotypes of NILs and their parental varieties were taken just before harvest. **a**, IR72 (*indica*) and IR72-NIL. **b**, Milyang21 (M21; *indica*) and M21-NIL. **c**, Milyang23 (M23; *indica*) and M23-NIL. **d**, Junam (JN; *japonica*) and JN-NIL.

After harvest, we analyzed the total grain yields of NILs (kg/10a). Compared to their parental cultivars, the grain yields of the IR72-NIL, Milyang21-NIL, and Milyang23-NIL were 850, 794, and 896 kg/10a, which corresponds to an increase of 5.1, 9.9, and 4.8 %, respectively in our experimental field condition (Response Figure 2). Previously, we showed that the grain yields per plant of NILs harboring *japonica* *OsSGR* allele increased to 10.6, 12.7, and 12.0 %, respectively, as compared to their parental cultivars. The differences in yield increase between our previous data and actual yield data in this year may be caused by the different cultivation year and the effects of three typhoons during the grain-filling period in this year. The productivity of JN-NIL harboring *indica* *OsSGR* allele was 739 kg/10a, which corresponds to the decrease of 5.6 %, as compared to parental *japonica* (Response Figure 2). We included the above results from the field test of NILs in 2019 as Supplementary Figure 18.

Response Figure 2. Actual grain yields of NILs grown in the field. IR72 (*indica*), IR72-NIL, Milyang21 (M21; *indica*), M21-NIL, Milyang23 (M23; *indica*), M23-NIL, Junam (JN; *japonica*) and JN-NIL. Data are means \pm SE ($n = 3$).

Minor point

1. Authors should add the statistical analysis in suppl. Fig. 3 and Table 4.

-Response: We performed the statistical analysis and revised the text in supplementary Figure 3 and Table 4.

Reviewer #3 (Remarks to the Author):

I accept the replies to my comments on the first version of this manuscript. However, I'm disappointed that the style of writing has not been changed throughout the text.

-Response: Many thanks to reviewer #3.

I hence repeat: Please avoid the use of the first person (I, we) and write the complete manuscript in the third person (e.g. it has been shown...). The first person is only appropriate for beliefs and convictions and is usually used in autobiographies. The description of scientific results rather requires impersonal reporting using third person pronouns instead of “we”, because the results hopefully exist without the reflections of the writing persons. In the abstract in lines 23, 25, 31 and later in line 67 etc. the personal style has been used.

-Response: We revised the text throughout the manuscript according to the reviewer’s suggestion.

Reviewer #4 (Remarks to the Author):

The authors have nicely addressed my concern about OsSGR being functional in regulating the senescence program (not just chlorophyll degradation) by showing that the indica NILs having japonica OsSGR allele maintained their photosynthetic capability longer than their parental varieties, and japonica NIL harboring indica OsSGR allele displayed decreased photosynthetic capability. They also analyzed the expression of the senescence marker gene OsNAP in rice plants. Activation lines showing early senescence accumulated higher levels of OsNAP transcripts in flag leaves while delayed leaf senescence mutants including *ossgr*, CRISPR/Cas9-genome editing and RNAi-suppressed lines displayed the decreased expression of OsNAP. The expression of OsNAP in indica (IR72)-NIL harboring japonica OsSGR allele was lower, as compared to IR72, and its expression was higher in the japonica-NIL having indica OsSGR allele compared to parental japonica variety.

-Response: We thank for this encouraging comments.

My second concern, about the discrepancy between OsSGR being a functional senescence regulator in this study and the results from earlier studies indicating that *sgr* is a nonfunctional stay-green locus (Park et al., 2007, Plant Cell; Sato et al., 2007, PNAS), is however not addressed. In Line 252-258, the authors stated that “The japonica OsSGR allele does not lead to an over stay-green but shows difference in its induction kinetics and induction level compared to the indica allele (Fig 3a). The japonica allele appears to exhibit a capacity to balance the degree of stay-green trait, photosynthetic competence, nutrient remobilization, when introduced into indica varieties, to lead to increased grain filling and productivity during the senescence period. Thus, NILs-OsSGR in the indica background harboring a japonica OsSGR allele exhibit the balanced stay-green with a delay of senescence, but unlike the stay-green knock out mutants that exhibit unbalanced and non-functional stay-green phenotype”. This statement is not convincing to me because based on the new data presented in Supplementary Figure 6 and Supplementary Figure 7, knockout mutants *ossgr* generated via CRISPR/Cas9 genome editing did show significant decrease in expression of the senescence marker gene OsNAP, implying that OsSGR is a regulator of senescence and loss-of-function in OsSGR did case delayed leaf senescence and theoretically increased photosynthetic capability and yield in this study. I suggest the authors to address this question seriously because *sgr* being a nonfunctional stay-green locus has been reported in several articles in multiple plant species.

-Response: We understand the reviewer’s concern.

OsSGR is a gene involved in chlorophyll degradation. The *japonica* alleles of *OsSGR* are progenitors of the *indica* alleles. The *indica* alleles lose chlorophyll earlier than *japonica* alleles due to natural variations in the promoter that activate the expression *OsSGR* earlier during the senescence period in our field condition in a temperate zone. Thus, the *indica* allele exhibits earlier senescence and accelerates the loss of photosynthetic capacity. In contrast, the *japonica* allele leads to typical slower senescence pattern observed for the *japonica* cultivars grown in temperate zones. Thus, when *japonica* alleles are introduced into *indica* cultivars, the NILs undergo the *japonica* type change of photosynthetic capacity during senescence period, which is slower than the parental *indica* cultivars.

To analyze the biochemical activity of *ossgr* allele, we conducted chlorophyll degradation activity assays *in vitro* and *in planta* to analyze the biochemical activity of the *ossgr* allele (Supplementary Fig. 11a, b). The results indicated that the *ossgr* allele did not have any chlorophyll breakdown activity, as compared to *japonica* and *indica* alleles. Therefore, *ossgr* mutants maintained high chlorophyll levels in senescent leaves, but mutant leaves did not maintain their photosynthetic activities longer than the WT leaves^{4,5}, leading to no yield advantage (Supplementary Fig. 6i-m). Knockout mutants *ossgr* in *indica* generated via CRISPR/Cas9 genome editing have no chlorophyll breakdown activity, just like *ossgr* mutants in *japonica*. Although chlorophyll breakdown in *ossgr* mutants is slower than that in WT leaves, previous reports have shown that *ossgr* mutants did not maintain photosynthesis competence, leading to a non-functional stay-green phenotype^{13,24}. *OsNAP*, a senescence marker gene, is increased in both WT and *ossgr* mutants, indicating that *ossgr* mutant plants undergo senescence despite of their stay-green phenotype. Thus, *ossgr* is a non-functional stay-green gene. However, the expression of *OsNAP* in *ossgr* mutants was not induced to WT levels. We suggest that the *OsSGR* gene itself or the higher level of chlorophyll in *ossgr* mutants may affect the expression of *OsNAP*. On the contrary, *indica*-NILs containing *japonica* *OsSGR* alleles maintain chlorophyll breakdown activity, which is lower than their parental *indica* cultivars, due to decreased expression of *OsSGR* during senescence. Thus, these NILs underwent slower chlorophyll loss and displayed prolonged photosynthetic abilities with slow senescence phenotypes similar to *japonica* cultivars, leading to a yield advantage. We revised the text in the ‘Discussion’ (page11~12, line 268~285).

References

1. Al-Tamimi, N. et al. Salinity tolerance loci revealed in rice using high-throughput non-invasive phenotyping. *Nat Commun* **7**, 13342 (2016).

2. Rice Knowledge Management Portal: <http://www.rkmp.co.in/>
3. Ookawa, T. et al. New approach for rice improvement using a pleiotropic QTL gene for lodging resistance and yield. *Nat Commun.* **1**, 132 (2010).
4. Park, S. Y. et al. The senescence-induced Staygreen protein regulates chlorophyll degradation. *Plant Cell* **19**, 1649-1664 (2007).
5. Sato, Y. Morita, R. Nishimura, M. Yamaguchi, H. & Kusaba, M. Mendel's green cotyledon gene encodes a positive regulator of the chlorophyll-degrading pathway. *Proc. Natl Acad. Sci. USA* **104**, 14169-14174 (2007).

Reviewers' comments:

Reviewer #2 (Remarks to the Author):

In this report, Shin D and his colleague indicated two important point, that natural variation in OsSGR promoter region might determine the lifespan variations between indica and japonica rice and the modify in this region could delay the senescence and enhance grain yield. In previous review, I pointed out that authors should analyze the photosynthetic ability of flag leaves or a canopy through growth stage and add the data of growth analysis (NAR) to clarify factors of yield characteristics. In new version, authors added the new data about net photosynthetic ratio (Fig. 4k) and RGR data (supplemental Fig.17). I am disappointed that this report is not suitable for Nature Communication, the top scientific journal because another group published the similar results about the delayed leaf senescence caused higher yield (Ramkumar et al., *Plants* 2019, 8, 375, <https://doi.org/10.3390/plants8100375>).

Major point

1. Ramkumar MK and his colleague reported with a novel stay-green mutant the delay of leaf senescence could improve the photosynthetic ability and yield. A stay-green locus they identified (SGM-3; on chr.9) was overlapped with QTL mapped by Shin et al.. And Ramkumar's group analyzed function the photosynthetic ability, chlorophyll contents and yield under normal or drought condition. Additionally, they indicated that OsSGR was a strong candidate gene of this locus and its different expression determined the allele.
2. In new version, authors added the data of net photosynthetic ratio between IR72 and NIL (Fig.4k) but the ratio of both plants was too low, for example, that of NIL was 8 or less and 5 or less in control at 5 weeks after heading. Generally, in same phase the photosynthetic ratio of rice flag leaf is more than 15 ($\mu\text{mol CO}_2/\text{m}^2/\text{s}$) in many reports (e.g., Jiang et al., *Jpn. J. Crop Sci*: 139-145, 1988). Authors should check the data.
3. Another important point of this paper is that the promoter polymorphisms in OsSGR affected the period of senescence. Authors should analyze the motif in the promoter region.
4. Authors indicated that variations in OsSGR controlled life span. However, heading dates were same among two japonica and indica rice plants (Supplementary Fig.1). These results showed these variations did not affect the life span but senescence period after heading.

Reviewer #4 (Remarks to the Author):

The authors stated in the text "OsNAP, a senescence marker gene, is increased in both WT and ossgr mutants, indicating that ossgr mutant plants undergo senescence despite of their stay-green phenotype. Thus, ossgr is a non-functional staygreen gene. However, the expression of OsNAP in ossgr mutants was not induced to WT levels" (Line 277-280).

In Supplementary Figure 6 d, the OSNAP levels, although increased during senescence, are significantly lower than that of WT, which means that senescence is significantly delayed. This does not support the conclusion "ossgr is a non-functional staygreen gene". This above mentioned statement does not explain why the ossgr mutants showed "no yield advantage (Supplementary Fig. 6i-m) ", and why the NILs (for which the authors did not provide OsNAP expression data) "underwent slower chlorophyll loss and displayed prolonged photosynthetic abilities with slow senescence phenotypes similar to japonica cultivars, leading to a yield advantage".

Since in the case of the OsSGR gene, chlorophyll content can no longer represent the progress of senescence, the authors should provide OsNAP expression and Fv/Fm data for all the critical lines to claim "senescence" phenotypes. A logical explanation of why senescence in both ossgr and NILs plants are delayed but only the NILs showed a yield advantage is lacking.

Point-by-point responses to the reviewers' comments.

1) Responses to Reviewer #2's comments:

Reviewer #2's comments.

In this report, Shin D and his colleague indicated two important point, that natural variation in OsSGR promoter region might determine the lifespan variations between indica and japonica rice and the modify in this region could delay the senescence and enhance grain yield. In previous review, I pointed out that authors should analyze the photosynthetic ability of flag leaves or a canopy through growth stage and add the data of growth analysis (NAR) to clarify factors of yield characteristics. In new version, authors added the new data about net photosynthetic ratio (Fig. 4k) and RGR data (supplemental Fig.17). I am disappointed that this report is not suitable for Nature Communication, the top scientific journal because another group published the similar results about the delayed leaf senescence caused higher yield (Ramkumar et al.,Plants 2019, 8, 375, <https://doi.org/10.3390/plants8100375>).

Major point

1. Ramkumar MK and his colleague reported with a novel stay-green mutant the delay of leaf senescence could improve the photosynthetic ability and yield. A stay-green locus they identified (SGM-3; on chr.9) was overlapped with QTL mapped by Shin et al. And Ramkumar's group analyzed function the photosynthetic ability, chlorophyll contents and yield under normal or drought condition. Additionally, they indicated that OsSGR was a strong candidate gene of this locus and its different expression determined the allele.

Responses.

We appreciate the reviewer for bring up this reference paper. The paper mentioned by the reviewer is a report on three stay-green mutants¹. The authors reported that one of the mutants, SGM-3, can improve harvest index and drought tolerance. Reviewer pointed out that the locus of SGM-3 mutant generated by EMS mutagenesis was overlapped with QTL (chr.9) mapped in our study and therefore *OsSGR* is a strong candidate gene of SGM-3 mutant. We carefully examined the article. We understand the paper is somewhat sophisticated but there is a discrepancy between the argument by the reviewer and the description in the paper.

The reviewer mentioned that "A stay-green locus they identified (SGM-3; on chr.9) was overlapped with QTL mapped by Shin et al. and Ramkumar's group analyzed function the photosynthetic ability,

chlorophyll contents and yield under normal or drought condition. Additionally, they indicated that OsSGR was a strong candidate gene of this locus and its different expression determined the allele.

However, the paper mentioned that, at the end of the Abstract, “Analysis of the earlier reported Quantitative Trait Loci (QTL) regions in SGM-3 revealed negligible variations from WT, suggesting it to be a novel SG mutant’. The paper also mentioned that in the Discussion part “suggesting that SGM-3 could be a novel and functional SG mutant (Page 12, line 43)”.

Thus, the authors of the paper have emphasized that the SGM-3 mutation is a novel one. The novelty of SGM-3 mutants in the paper was argued based on the following data.

There were no mutations on *OsSGR* gene in SGM-3, which was described in the article (page 11, line 8-10). The results suggested that SGM-3 is not caused by the *OsSGR* mutation.

They analyzed the QTL region on chromosome 9 of the functional SG mutant, SNU-SG1⁶, but results showed no significant changes from WT. (Page 12, line 40-42). Therefore, they suggested that SGM-3 could be the novel functional SG mutant and will map the causal gene for future works.

The authors mentioned that “Thus, SGM-1 and SGM-2 were inferred as cosmetic SG mutants while SGM-3 as a functional SG mutant. (Page 11, line 28-29). As *ossgr* mutant is a non-functional, SGM-3 is different from *OsSGR*.

In addition, the expression of *OsSGR* in SGM-3 was higher than that of WT (Figure 3; Figure 7), although SGM-3 showed the delayed senescence (Figure 1). The authors mentioned that “SGM-3 also had the highest upregulation for NOL, SGR and PAO” (Page 10, line 7-8). Higher expression of *OsSGR* should lead to earlier loss of greenness but this was not the case in SGM-3, indicating the difference between SGM-3 and *ossgr* mutants.

Yet, we appreciate the value of the reference and added the following sentence in the discussion part of the revised manuscript (Page 13, line 313-316).

In agreement with our observation, a recent report showed that an extended photosynthetic competence during senescence stage leads to increased harvest index in SGM-3 mutant of upland rice variety Nagina 22; SGM-3 was suggested to be a “novel and functional stay green mutant”³¹.

2. In new version, authors added the data of net photosynthetic ratio between IR72 and NIL (Fig.4k) but

the ratio of both plants was too low, for example, that of NIL was 8 or less and 5 or less in control at 5 weeks after heading. Generally, in same phase the photosynthetic ratio of rice flag leaf is more than 15 ($\mu\text{mol CO}_2/\text{m}^2/\text{s}$) in many reports (e.g., Jiang et al., Jpn. J. Crop Sci: 139-145, 1988). Authors should check the data.

Response.

We thank the reviewer for raising this concern. According to the reviewer's suggestion, we checked the article by Jiang et al. (1988)⁷. They used different unit from ours as below.

Table 1. Comparison of the maximum photosynthetic rate ($\text{mgCO}_2/\text{dm}^2/\text{hr}$) in a day between Nipponbare and Akenohoshi at different growth stages.

Leaf position	Variety	Maximum tillering stage	Panicle formation stage	Ripening stage		
				Early	Middle	Late
12 th	Nipponbare	35.4 \pm 3.4 (100)*	21.8 \pm 2.9 (61.6)	—	—	—
	Akenohoshi	37.6 \pm 2.3 (100)	25.5 \pm 2.1 (67.8)	—	—	—
14 th	Nipponbare	—	32.7 \pm 2.9 (100)	20.4 \pm 3.2 (62.4)	14.6 \pm 2.0 (44.6)	—
	Akenohoshi	—	32.6 \pm 1.6 (100)	21.4 \pm 1.0 (65.6)	20.2 \pm 1.6 (62.0)	—
16 th (flag leaf)	Nipponbare	—	—	28.2 \pm 1.7 (100)	25.4 \pm 1.3 (90.1)	16.6 \pm 2.7 (58.9)
	Akenohoshi	—	—	31.7 \pm 1.8 (100)	31.5 \pm 2.5 (99.4)	21.1 \pm 2.0 (66.6)

* Values in parentheses show the percentage of the photosynthetic rates of 12th, 14th and 16th leaves to those at the maximum tillering, panicle formation and early ripening stages, respectively.

They used the unit of $\text{mgCO}_2/\text{dm}^2/\text{hr}$. We used the unit of $\mu\text{molCO}_2 \text{m}^{-2} \text{S}^{-1}$ as below.

Fig. 4k. The analysis of the net CO₂ assimilation rate in flag leaves of IR72 and IR72-NIL during the grain-filling stage (4, 5, 6 and 7 weeks after heading). Data are means \pm SE (n = 5).

Supplementary Figure 17h. The analysis of the net CO₂ assimilation rate in flag leaves of JN and JN-NIL during the grain-filling stage (4, 5, 6 and 7 weeks after heading). Data are means \pm SE (n = 5).

Thus, a direct numerical comparison may not be appropriate. Besides, the text in the paper was in Japanese, which made it difficult for us to understand the details of the instrument and methods such as the exact timing (e.g., days after heading) of the three time points during ripening stage.

However, to answer to the reviewer's concern, we compared our data with a few previous reports, where a same unit, and a same or similar instrument and time points were used. We measured the net CO₂ assimilation rate from 4 weeks after heading. In the paper by Ramkumar et al, photosynthetic rate of WT (Nagina 22 (*aus*)) was 11 μmol CO₂ m⁻² s⁻¹ at 28 days after heading¹, which was comparable to our results (12.82 in IR72 (*indica*), Fig. 4k). Photosynthetic rates of two popular *japonica* cultivars, Takanari and Koshihikari, were about ~17 and ~12 μmol CO₂ m⁻² s⁻¹, respectively at 4.5 week after heading⁸, which were comparable to our results (~15 μmol CO₂ m⁻² s⁻¹ at 4 weeks after heading in Junam (*japonica*), Supplementary Figure 17h). At 30 days after heading, photosynthetic rates in Suwon490 (*japonica*), Andabyeo (*japonica*), SNU-SG1, and IR71451-40 were ~9, ~7, ~ 15, and ~ 10 μmol m⁻² s⁻¹, respectively⁹, which were comparable to our results. At 25 days after heading, a late-stage vigor super rice cultivar (Y-liangyou 087) and an elite rice cultivar (Teyou 838) showed ~16 and ~15 μmol CO₂ m⁻² s⁻¹, respectively¹⁰.

We understand that the exact numerical values of photosynthetic rates may be variable depending on rice varieties and growth conditions. Yet, the comparison of our data with the previous data as mentioned above shows that our data is in a comparable range. We did not include this response in the revised manuscript, as our values were not peculiar.

3. Another important point of this paper is that the promoter polymorphisms in OsSGR affected the period of senescence. Authors should analyze the motif in the promoter region.

Response.

We understand the reviewer's point. We added the following paragraph in the Discussion part of the revised manuscript (Page 11, line 256-267)¹¹.

When we analyzed the binding motifs for transcriptional factors in the 2-kb *OsSGR* promoter region using NEW PLACE (A Database of Plant Cis-acting Regulatory DNA element; <https://www.dna.affrc.go.jp/PLACE/?action=newplace>), we found that the *japonica* and *indica* promoters contained same numbers (thirteen) of WRKY binding motifs and no NAC binding motif. However, there is a distinctive difference in the Dof binding motifs between the two promoters. There are fourteen Dof

protein binding motif (consensus AAAG sequences or its reversibly complementary sequence, CTTT³⁰) in the *japonica* promoter. On the other hand, in the *indica* promoter, there is a new Dof protein binding motif formed by insertion of AAAAGCTC (position -1, 377; Supplementary Fig. 9). This region with the new Dof binding motif is tightly associated with low levels of chlorophyll contents. We anticipate that this new Dof binding motif and the respective Dof transcription factor may together lead to the *indica* type phenotype of chlorophyll loss. Elucidating the molecular mechanisms regulating the early and higher induction of *OsSGR* through evolution of the promoter region should be a key future effort.

4. Authors indicated that variations in *OsSGR* controlled life span. However, heading dates were same among two *japonica* and *indica* rice plants (Supplementary Fig.1). These results showed these variations did not affect the life span but senescence period after heading.

Response.

We appreciate the reviewer's point. We added the following paragraph in Discussion part in response to the reviewer's point. We like to note that heading dates of cultivars do not necessarily reflect their respective lifespans, as cultivars show differences in senescence process after heading with the concomitant differences in their lifespans (Page 13/14, line 317-326)¹².

Lifespan indicates the maximal life expectancy from the seed to seed and the length of time that plants live or expected to live³². In this regard, the lifespan of rice plants is the duration time from seed germination to the panicle senescence and death associated with grain maturation. In analyzing senescence processes of rice plants in our experiments, we chose two *japonica* and *indica* rice cultivars, which showed the same heading date (Supplementary Fig. 1) to avoid an influence of the differences in reproductive timing on senescence processes and the related lifespan. Despite of the same heading dates, the panicles of the two *indica* cultivars showed earlier senescence than those of the two *japonica* cultivars (Fig. 1h), as quantified by colorimetric assays (Fig. 1i). Thus, the lifespans of these rice cultivars were largely related to senescence process of the panicles after heading, which is controlled by the differential expression levels and kinetics of *OsSGR* (Fig. 4c, f, h, j).

2) Responses to Reviewer #4's comments.

Reviewer #4's comments.

The authors stated in the text “OsNAP, a senescence marker gene, is increased in both WT and ossgr mutants, indicating that ossgr mutant plants undergo senescence despite of their stay-green phenotype. Thus, ossgr is a non-functional staygreen gene. However, the expression of OsNAP in ossgr mutants was not induced to WT levels” (Line 277-280).

In Supplementary Figure 6d, the OSNAP levels, although increased during senescence, are significantly lower than that of WT, which means that senescence is significantly delayed. This does not support the conclusion “ossgr is a non-functional staygreen gene”. This above mentioned statement does not explain why the ossgr mutants showed “no yield advantage (Supplementary Fig. 6i-m)”, and why the NILs (for which the authors did not provide OsNAP expression data) “underwent slower chlorophyll loss and displayed prolonged photosynthetic abilities with slow senescence phenotypes similar to japonica cultivars, leading to a yield advantage”.

Since in the case of the OsSGR gene, chlorophyll content can no longer represent the progress of senescence, the authors should provide OsNAP expression and Fv/Fm data for all the critical lines to claim “senescence” phenotypes. A logical explanation of why senescence in both ossgr and NILs plants are delayed but only the NILs showed a yield advantage is lacking.

Response:

We fully understand the reviewer’s concern. We tried to make the confusion clearer in a few places in Result and Discussion.

The first point of the reviewer’ concerns is regarding the expression of *OsNAP* in the *OsSGR* mutant plants. The nature of *OsSGR* as a non-functional stay-green gene has been well described before². It encodes the enzyme Mg⁺⁺-dechelatase in chlorophyll degradation pathway and is not a regulatory gene⁴. However, it was known that the knock-out mutation in this enzyme leads to some degree of delayed senescence upon a prolonged growth as shown below in c². In this measurement, the Fv/Fm value was higher in 30 days after heading in the mutant than in wild type.

The data on the above figure is consistent with our observation that expression of *OsNAP* in *ossgr* mutant plants is not induced to the wild type level during senescence (Supplementary Figures 6d and 7c), as pointed out by the reviewer as below.

“In Supplementary Figure 6 d, the OSNAP levels, although increased during senescence, are significantly lower than that of WT, which means that senescence is significantly delayed”.

We do not know the mechanism underlying the partial delay of some senescence symptoms such as Fv/Fm and expression of *OsNAP* in the *ossgr* mutant after a prolonged growth. Perhaps, high level of the remaining chlorophyll may have some effect on the partial delay of senescence. However, in the above data, it is also noted that the net photosynthesis (Pn) is not maintained even after 30 days after heading, showing the characteristics of non-functional stay-green mutants².

We added the following paragraph to respond to the reviewer’s point (Page 12, line 276-292). In addition, we made some minor changes in Result section (in red in Page 6) to respond to the reviewer’s concern in the expression of *OsNAP*^{2,3,4,5}.

The nature of *OsSGR* as a non-functional stay-green gene has been well characterized previously^{13,15}. *OsSGR* encodes the enzyme Mg^{++} -dechelatase in the chlorophyll degradation pathway and is not a regulatory gene¹⁹. We conducted chlorophyll degradation activity assays *in vitro* and *in planta* to analyze the biochemical activity of the proteins derived from various *ossgr* alleles (Supplementary Fig. 11a, b). The results showed that the protein from the *ossgr* knockout mutant allele showed a negligible

enzyme activity toward chlorophyll degradation. Accordingly, *ossgr* mutant plants maintain a high level of chlorophyll content in senescent leaves but without comparable maintenance of net photosynthesis¹⁵, which is a characteristic feature of non-functional stay-green mutants. However, the *ossgr* knockout mutation leads to delay in some aspects of functional senescence upon a prolonged growth; Fv/Fm value was higher in 30 days after heading in the mutant plants than in wild type plants¹⁵. This previous observation is consistent with the expression pattern of *OsNAP*²⁴, a senescence marker gene, in *ossgr* mutant plants. Expression of *OsNAP* is increased in both wild type and *ossgr* mutant plants during the grain filling stage, indicating that *ossgr* mutant plants undergo senescence despite of their stay-green phenotype. However, the expression of *OsNAP* was lower in *ossgr* knockout mutants than in wild type plants, indicating a partial delay of a functional senescence in terms of *OsNAP* expression (Supplementary Figures 6d, 7c) in addition to the Fv/Fm values¹⁵

We also added the paragraph below to respond another point of the reviewer which is “A logical explanation of why senescence in both *ossgr* and NILs plants are delayed but only the NILs showed a yield advantage is lacking” (Page 12/13, line 293-316)¹.

Here, we showed that rice grain yield can be increased by replacing the *indica* allele of *OsSGR* with the *japonica* allele. As *OsSGR* is a non-functional stay-green gene in terms of net photosynthesis, it seems paradoxical to increase rice yield via *OsSGR*. It was the difference in the promoter region of the *indica* and *japonica* alleles of the *OsSGR* gene that led to yield increase in the NILs we generated. Unlike the *ossgr* knockout mutation with no enzyme activity of OsSGR, the proteins from *japonica* and *indica* alleles showed comparable enzyme activity (Supplementary Figure 11). However, the promoter regions of the two alleles of *OsSGR* are diverged (Supplementary Figure 9). Compared to the promoter of the *japonica* allele, the promoter of the *indica* allele led to earlier and higher induction of *OsSGR* (Fig. 3a) and to earlier loss of chlorophyll (Fig. 1) with concomitant reduction of photosynthesis (Fig. 4k) and grain yield (Fig. 4l). The senescence response in rice plants with the *indica* allele is similar to that in the activation tagging lines (Supplementary Figure 5) with the *japonica* alleles, where *OsSGR* expression is increased at a later stage with lower level of chlorophyll and reduced grain yield (Supplementary Figure 5). In generating the NIL lines with increased yield, we replaced the *indica* allele with the *japonica* allele, so that the induction of *OsSGR* is slower than the parental lines with the *indica* allele (Fig. 4j). This leads to slower loss of chlorophyll (Fig. 4g, i), higher photosynthesis (Fig. 4k), and increased yield (Fig. 4l) compared to those in their parental lines. Thus, the senescence response of rice plants with the *japonica* allele with slower induction of *OsSGR* became comparable to a functional stay-green phenotype, unlike the *ossgr* knock-out mutant which showed non-functional stay-green phenotype with no yield advantage.

As the heading date of these lines are same, our results show that a senescence period with extended photosynthetic competence can lead to higher productivity. In agreement with this notion, a recent report showed that an extended photosynthetic competence during senescence stage leads to increased harvest index in SGM-3 mutant of upland rice variety Nagina 22; SGM-3 was suggested to be a “novel and functional stay green mutant”³¹.

We also like to note that we measured the expression of *OsNAP* and Fv/Fm values of IR72, IR72-NIL, JN and JN-NIL during senescence stage to follow the senescence state in these lines, which were already included in the manuscript (Supplementary Figure 17).

References

1. Ramkumar, M et al. A novel stay-green mutant of rice with delayed leaf senescence and better harvest index confers drought tolerance. *Plants* **8**, 375 (2019).
2. Jiang, H. et al., Molecular cloning and function analysis of the *stay green* gene in rice. *Plant J.* **52**, 197–209 (2007).
3. Park, S. Y. et al. The senescence-induced staygreen protein regulates chlorophyll degradation. *Plant Cell* **19**, 1649-1664 (2007).
4. Shimoda, Y. Ito, H. & Tanaka, A. Arabidopsis STAY-GREEN, Mendel's Green Cotyledon Gene, Encodes Magnesium-Dechelatase. *Plant Cell* **28**, 2147-2160 (2016).

5. Liang, C. et al. OsNAP connects abscisic acid and leaf senescence by fine-tuning abscisic acid biosynthesis and directly targeting senescence-associated genes in rice. *Proc. Natl Acad. Sci. USA* **111**, 10013-10018 (2014).
6. Yoo, SC et al. Quantitative trait loci associated with functional stay-green SNU-SG1 in rice. *Mol Cells*. **24**, 83-94 (2007).
7. Jiang, C. Hirasawa, T. & Ishihara, K. Physiological and ecological characteristics of high yielding varieties in rice plants. *Jpn. J. Crop Sci.* **57**, 139-145(1988).
8. Taylaran, R.D. Adachi, S. Ookawa, T. Usuda, H. & Hirasawa, T. Hydraulic conductance as well as nitrogen accumulation plays a role in the higher rate of leaf photosynthesis of the most productive variety of rice in Japan. *J Exp Bot.* **62**, 4067-4077 (2011).
9. Fu, J.D. & Lee, B.W. Changes in photosynthetic characteristics during grain filling of functional stay-green rice SNU-SG1 and its F1 hybrids. *J. Crop Sci. Biotech.* **11**, 75-82 (2008).
10. Huang, M. Chen, J. Cao, F. Jiang, L. & Zou, Y. Root Morphology Was Improved in a Late-Stage Vigor Super Rice Cultivar. *PLoS One.* **10**, e0142977 (2015).
11. Noguero, M. Atif, R.M. Ochatt, S. & Thompson, R.D. The role of the DNA-binding One Zinc Finger (DOF) transcription factor family in plants. *Plant Sci.* **209**, 32-45 (2013).
12. Thomas, H. Senescence, ageing and death of the whole plant. *New Phytol.* **197**, 696-711 (2013).
13. Park, S. Y. et al. The senescence-induced staygreen protein regulates chlorophyll degradation. *Plant Cell* **19**, 1649-1664 (2007).
15. Jiang, H. et al., Molecular cloning and function analysis of the *stay green* gene in rice. *Plant J.* **52**, 197-209 (2007).
19. Shimoda, Y. Ito, H. & Tanaka, A. Arabidopsis STAY-GREEN, Mendel's Green Cotyledon Gene, Encodes Magnesium-Dechelataase. *Plant Cell* **28**, 2147-2160 (2016).

24. Liang, C. et al. OsNAP connects abscisic acid and leaf senescence by fine-tuning abscisic acid biosynthesis and directly targeting senescence-associated genes in rice. *Proc. Natl Acad. Sci. USA* **111**, 10013-10018 (2014).
30. Noguero, M. Atif, R.M. Ochatt, S. & Thompson, R.D. The role of the DNA-binding One Zinc Finger (DOF) transcription factor family in plants. *Plant Sci.* **209**, 32-45 (2013).
31. Ramkumar, M et al. A novel stay-green mutant of rice with delayed leaf senescence and better harvest index confers drought tolerance. *Plants* **8**, 375 (2019).
32. Thomas, H. Senescence, ageing and death of the whole plant. *New Phytol.* **197**, 696-711 (2013).